# Epigenetically distinct synaptic architecture in clonal compartments in the teleostean dorsal pallium

**Yasuko Isoe**[1,2,3]*[†], **Ryohei Nakamura**[2][†], **Shigenori Nonaka**[4,5,6], **Yasuhiro Kamei**[4,7], **Teruhiro Okuyama**[2,8], **Naoyuki Yamamoto**[9], **Hideaki Takeuchi**[2,3,10‡], **Hiroyuki Takeda**[2,11‡]

[1]Department of Molecular and Cellular Biology, Faculty of Arts and Sciences, Harvard University, Cambridge, United States; [2]Department of Biological Sciences, Graduate School of Science, The University of Tokyo, Tokyo, Japan; [3]Graduate School of Natural Science and Technology, Okayama University, Okayama, Japan; [4]Department of Basic Biology, Graduate School for Advanced Studies, Okazaki, Japan; [5]Spatiotemporal Regulations Group, Exploratory Research Center for Life and Living Systems, Okazaki, Japan; [6]Laboratory for Spatiotemporal Regulations, National Institute for Basic Biology, Okazaki, Japan; [7]Trans-Scale Biology Center, National Institute for Basic Biology, Okazaki, Japan; [8]Institute for Quantitative Biosciences (IQB), The University of Tokyo, Tokyo, Japan; [9]Department of Animal Sciences, Graduate School of Bioagricultural Sciences, Nagoya University, Nagoya, Japan; [10]Department of Integrative Life Sciences, Graduate School of Life Sciences, Tohoku University, Sendai, Japan; [11]Kyoto Sangyo University, Faculty of Life Sciences, Kamigamo Motoyama, Kyoto, Japan

***For correspondence:**
yasuko_isoe@fas.harvard.edu

[†]These authors contributed equally to this work
[‡]These authors also contributed equally to this work

**Competing interest:** The authors declare that no competing interests exist.

**Abstract** The dorsal telencephalon (i.e. the pallium) exhibits high anatomical diversity across vertebrate classes. The non-mammalian dorsal pallium accommodates various compartmentalized structures among species. The developmental, functional, and evolutional diversity of the dorsal pallium remain unillustrated. Here, we analyzed the structure and epigenetic landscapes of cell lineages in the telencephalon of medaka fish (*Oryzias latipes*) that possesses a clearly delineated dorsal pallium (Dd2). We found that pallial anatomical regions, including Dd2, are formed by mutually exclusive clonal units, and that each pallium compartment exhibits a distinct epigenetic landscape. In particular, Dd2 possesses a unique open chromatin pattern that preferentially targets synaptic genes. Indeed, Dd2 shows a high density of synapses. Finally, we identified several transcription factors as candidate regulators. Taken together, we suggest that cell lineages are the basic components for the functional regionalization in the pallial anatomical compartments and that their changes have been the driving force for evolutionary diversity.

## Editor's evaluation

This important article highlights the clonal organization of the dorsal telencephalon, a major region of the vertebrate brain. The authors' analyses reveal a distinct gene expression and provide a high-quality chromatin accessibility map of the adult teleost fish medaka. In addition, synaptic genes have a distinct chromatin landscape and expression pattern from one of the regions of the dorsal pallium, aiming to describe an evolutionary origin for these neurons.

## Introduction

The telencephalon is an essential brain part for an animal's cognitive functions and is highly diverse in structure among vertebrates. The dorsal telencephalon, or the pallium, is divided into several distinct regions in mammals; for example, the cerebral neocortex in the mammalian dorsal pallium (MDP), the hippocampus in the mammalian medial pallium (MMP), and the basolateral amygdala in the mammalian lateral pallium (MLP) (*Briscoe and Ragsdale, 2019*; *Yamamoto et al., 2007*; *Northcutt, 2011*; *Yamamoto et al., 2017*). The MDP is characterized by a six-layered structure and by stereotypical projections from all sensory modalities. The non-mammalian dorsal pallium generally lacks a multilayered structure, with the exception of non-avian reptiles (*Aboitiz et al., 2002*; *Aboitiz et al., 2003*), but rather exhibits a compartmentalized architecture that is highly diverse across species. Little is known about how the anatomical diversity in the dorsal pallium has emerged during evolution. Nonetheless, the neocortex in mammals, as well as the dorsal pallium in non-mammalian clades, receives structured input from all sensory modalities, and these structures therefore are hypothesized to fulfill similar function. In teleost, the number of compartments in the dorsal pallium is also quite diverse among species. Somatosensory input into the dorsal pallium in marbled rockfish (*Yamamoto et al., 2007*; *Ito et al., 1986*) and visual projections from the optic tectum in the yellowfin goby (*Hagio et al., 2021*; *Hagio et al., 2018*; *Bloch et al., 2020*) are just a few specific examples of such sensory projections. However, because of the lack of genetic tools in these species, the molecular characterization of the dorsal pallium in teleosts is still largely underexplored. In zebrafish (*Danio rerio*), the most popular fish model organism, a clear dorsal pallium could not be delineated (*Porter and Mueller, 2020*; *Mueller et al., 2011*), which has made a detailed characterization difficult in this species.

To study how the specific architecture of the dorsal pallium in teleosts emerges and is maintained throughout life, we combined two orthogonal approaches. First, cell lineage analysis was used to visualize cellular subpopulations derived from the same neural stem cells at an early developmental stage. Since post-hatch neurogenesis occurs throughout life in all teleosts, such cell lineage analysis of neural progenitors allows us to collectively label individual neural progenitors populations that may serve similar functions (*Raj et al., 2018*; *Furlan et al., 2017*; *Dirian et al., 2014*). Second, for characterizing gene expression within these clonal units, we focused on epigenetic regulation of transcription, where ATAC-sequencing (*Letelier, 2018*; *Yin et al., 2020*) allows for the identification of specifically regulated genes within each population, whose expression levels are further quantified by subsequent RNA-sequencing.

Here, we selected medaka fish (*Oryzias latipes*) as a model organism for three reasons. First, medaka possesses a dorsal pallium (Dd2) that is demarcated by a cellular boundary, which makes it anatomically distinct (*Anken and Bourrat, 1998*) and facilitates the isolation of this particular region. Second, transgenic medaka lines are available to visualize cell lineages in the brain (*Okuyama et al., 2013*). Third, the genome size of medaka is smaller than that of zebrafish (*Kasahara et al., 2007*), the genome sequence is of high quality (*Ichikawa et al., 2017*), and it is readily available for epigenetic analysis (*Nakamura et al., 2021*; *Cheung et al., 2017*).

We first investigated the clonal architecture of the entire telencephalon in medaka and found that all anatomical regions within the pallium, including Dd2, are formed by mutually exclusive clonal units. We next investigated that clonal units in Dd2 in particular possess a unique open chromatin landscape, where synaptic genes were actively regulated. Subsequently, we examined the transcriptional regulatory mechanism in Dd2 to identify some candidate transcription factors (TFs) that contribute to the unique epigenetic landscape in the dorsal pallium. Our data suggest that clonal units in the pallium show modular properties, and in particular, the dorsal pallium is constructed with distinct synaptic architecture according to different rules from those of other pallial regions.

## Results

### Anatomical regions in the adult medaka pallium consist of cell bodies of cell lineages

Other than in mammals, the teleost brain grows continuously throughout life via post-hatch neurogenesis (*Kuroyanagi et al., 2010*; *Figure 1A*, left), such that the anatomical classification framework needs to be adapted to this expansion. For example, the nomenclature of the adult medaka telencephalon has been complicated by the difficulties in identifying the small brain structures that emerge in adult

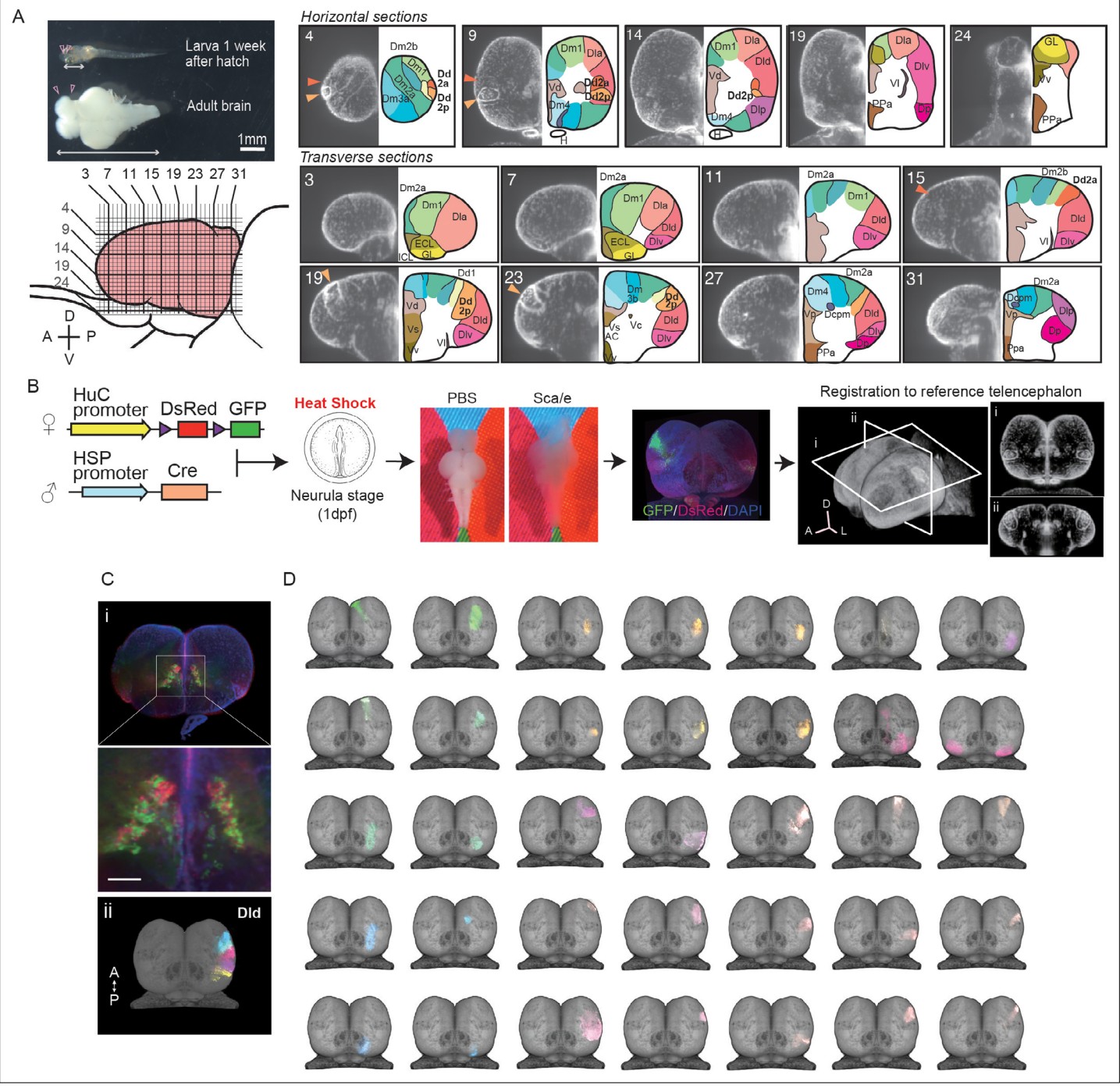

**Figure 1.** Clonal architecture in the adult medaka telencephalon. (**A**) A larval medaka fish (1 week day post fertilization) and a dissected adult brain (top left). Pink triangles indicate the position of the telencephalon in the larval and adult brain. The white two-way arrows indicate the length of the whole larval brain. Scale bar: 1 mm. Schematic drawing of the lateral view of the adult telencephalon (pink, bottom left). Horizontal and vertical lines indicate the position of optical sections in the brain atlas on the right panel. Redefined anatomical regions of adult medaka telencephalon are shown (right). Optical horizontal sections (top right) and transverse sections (bottom right). Orange triangles indicate the position of the dorsal pallial regions we focus on (dark orange: Dd2a; light orange: Dd2p). For each section, the left picture shows DAPI signals and the right shows the brain atlas. (**B**) Experimental procedure to label cell lineages. Cre-loxp recombination was induced by a short heat shock at the neurula stage of transgenic embryos (Tg (HuC:loxp-DsRed-loxp-GFP) × Tg(HSP:Cre)). Dissected brains were stained with DAPI, cleared in Sca*l*e solution, and light-sheet microscopy images were taken. Fluorescent signals were registered to a reference telencephalon. Optical horizontal and transverse sections are shown (**i, ii**). A: anterior direction; D: dorsal direction; L: lateral direction. (**C**) Examples of anatomical regions consist of clonally related cells. Multiple cell lineages mix and constitute the subpallial regions, for example, the dorsal medial part of the subpallium (Vd) (an example image from a single fish's optical section is shown [**i**]), while

*Figure 1 continued on next page*

*Figure 1 continued*

the pallium consists of cell lineages in an exclusive mosaic pattern. The clonally related cells in the dorsal part of the lateral pallium (Dld) are shown as an example (**ii**). A: anterior; P: posterior direction. Scale bar: 100um. (**D**) Examples of the structure of cell lineages identified in the telencephalon. Dorsal views are shown. Colors of cell lineages indicate the position of cell somas in the anatomical region in (**A**). Dc: the dorso-central telencephalon; Dcpm: the posterior medial nucleus of the dorso-central telencephalon; Dm: the medial part of the dorsal pallium; Dl: the dorsal lateral pallium; Dla: the anterior part of the dorso-lateral telencephalon; Dld: the dorsal part of the middle and posterior part of the dorso-lateral telencephalon; Dlv: the ventral part of the middle and posterior part of the dorso-lateral telencephalon; Dlp: the posterior regions of the dorso-lateral telencephalon; Dd: the dorso-dorsal telencephalon; Dp: the posterior part of dorsal telencephalon; Vd: the dorsal medial part of the subpallium; Vs: the supracommissural part of ventral telencephalon; Vv: the ventral part of the subpallium; ECL: the external layer of the olfactory bulb; ICL: the internal layer of the olfactory bulb.

The online version of this article includes the following figure supplement(s) for figure 1:

**Figure supplement 1.** Details of redefined atlas of the telencephalon.

**Figure supplement 2.** Expression of cell-type marker genes in the adult medaka telencephalon.

**Figure supplement 3.** Statistics of clonal units.

**Figure supplement 4.** Radial glial projection in the adult medaka telencephalon.

brain sections throughout development (*Anken and Bourrat, 1998*; *Ishikawa et al., 1999*). Here, we first redefined the anatomical regions in the medaka telencephalon based on the three-dimensional distribution of cell nuclei by DAPI staining and tissue clearing (*Figure 1A*, right, *Figure 1—figure supplement 1*, *Video 1*). To acquire optical sections of the whole adult telencephalon, we applied a tissue clearing solution, Sca*l*eA2 (*Hama et al., 2011*,) followed by light-sheet microscopic imaging. We also defined the anatomical regions by immunostaining of cell-type-specific neural markers, such as CaMK2α, parvalbumin, and GAD65/67 (*Table 1*, *Figure 1—figure supplement 2*). Here, we redefined the medial part of the pallial region (Dm) into several subcompartments (Dm2a, Dm2b, Dm3a, and Dm3b) by clear DAPI-positive boundaries. These subcompartments are absent in previous brain atlases. Also, we found clear anatomical boundaries in the dorsal pallium where different marker genes are expressed along the anterior–posterior axis (Dd2a, 2p) (*Figure 1A*, right, *Figure 1—figure supplement 2A*; a detailed description of nomenclature is provided in *Table 2*).

Next, to uncover the clonal architecture in the telencephalon, we genetically visualized the spatial distribution of cell lineages as previously established (*Okuyama et al., 2013*; *Figure 1B*). We used the transgenic line (Tg (HuC:loxp-DsRed-loxp-GFP)) that visualized neural progenitors by the HuC promoter, which labels 60–70% of all neurons in the adult telencephalon (*Figure 1—figure supplement 2B*). We crossed it to a second line (Tg (HSP-Cre)) that expresses Cre recombinase under the heat shock protein promoter. The Cre-loxp recombination was induced stochastically by heat shock at the neurula stage (stage 16–17; *Iwamatsu, 2004*) to visualize the progenitors derived from the same neural stem cells. We chose this developmental stage because the heat shock at an earlier developmental stage labeled to many neurons, and the heat shock at a later stage rarely induced recombina-

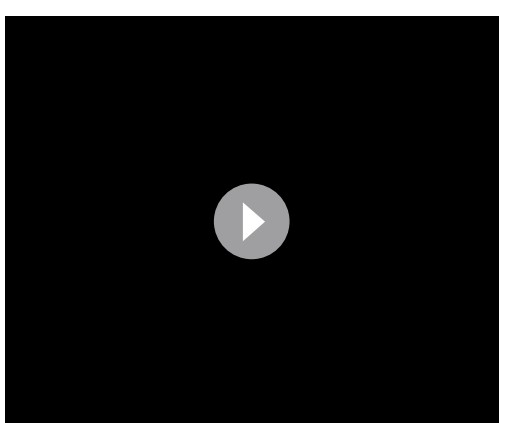

**Video 1.** Reference brain. DAPI-based reference brain. The slices show the horizontal view from the ventral to the dorsal direction.

https://elifesciences.org/articles/85093/figures#video1

tion in our Tg line, which is presumably because of the positional effect of the insertion site in this transgenic line. Then, we cleared the adult brains with Sca*l*e solution, stained them with DAPI, and performed light-sheet microscope imaging. We first found that GFP-signal distributed differently between the pallium and subpallium; GFP-positive cell somas mixed with DsRed-positive cell somas in the subpallial regions (*Figure 1Ci*), while GFP-positive cells in the pallium formed exclusive compartments (*Figure 1B*, *Videos 2 and 3*).

For further investigation of the clonal architecture in each anatomical region, we analyzed the structure of the clonally related cells by applying a normalization method (*Ostrovsky et al., 2013*; *Figure 1B*). To that end, we registered the telencephalon of 81 medaka where 523 GFP-positive subpopulations were detected in total to a reference telencephalon. GFP-positive subpopulations

**Table 1.** Summary of marker gene expression in the telencephalic regions in medaka. Several marker gene expressions were assessed with immunostainings (*Figure 1—figure supplement 2*).

| | CaMK2 | Parvalbumin | GAD65/67 | GAD67 |
|---|---|---|---|---|
| Dla | + | n.a. | n.a. | n.a. |
| Dld | +/− | + | + | ++ |
| Dlp | +/− | − | - | n.a. |
| Dlv | + | + | + | ++ |
| Dd1 | + | ++ | - | + |
| Dd2a | + | ++ | ++ | + |
| Dd2p | ++(ventral) | + | ++(dorsal) | + |
| Dm1 | − | − | + | n.a. |
| Dm2a | + | − | + | + |
| Dm2b | ++ | − | + | + |
| Dm3a | ++ | − | + | + |
| Dm3b | ++ | − | + | + |
| Dm4 | ++ | ++ | - | n.a. |
| Dp | ++ | + | - | n.a. |
| Dc | + | - | - | n.a. |
| Dcpm | − | - | n.a. | n.a. |
| GL | + | n.a. | ++ | + |
| ICL | − | ++ | n.a. | n.a. |
| Vv | − | ++ | - | - |
| Vd | − | ++ | - | - |
| Vp | ++ | + | - | - |
| Vs | − | + | - | - |
| Vl | − | ++ | - | - |

- signals not detected; +: signals detected; ++: signals strongly detected; n.a.: not analyzed.

that appeared multiple times in the same area were defined as clonally related units (*Figure 1D*). On average, we could visualize 6–8 units per fish (*Figure 1—figure supplement 3A–C*). We did not observe the GFP-positive units in a uniform frequency in the telencephalon, but more frequently in Dld, Dd2, and the subpallial regions than other areas (*Figure 1—figure supplement 3C*). We found the smallest clone in Dcpm, which included a few dozens of cells. We did not find any laterality of clonally related units, but we found more observed GFP-positive cells in the pallium than in the subpallium (*Figure 1—figure supplement 3B–E*). In the subpallium, we observed GFP-positive cells mixed with DsRed-positive cell with a 50% probability. On the other hand, most of the GFP-positive cells in the pallium were not mixed with DsRed-positive cells and did not locate across multiple regions (*Figure 1—figure supplement 3F*). For example, in the lateral pallium (Dl), the dorsal and ventral part of Dl were clearly different in terms of how the GFP-positive cell bodies are distributed (*Figure 2B*). Specifically, in the dorsal pallium, the anterior part (Dd2a) and the posterior part (Dd2p) consist of two clonally related units each that cluster laterally and medially, respectively. Taken together, we found that the anatomical regions in the pallium are formed by around 50 of the compartments of clonally related cell bodies, and the clonally related compartments distribute exclusively in one anatomical region but not across many anatomical regions.

**Table 2.** Nomenclature of the adult telencephalon.

For the medaka brain research, two brain atlases of adult medaka have been used (*Anken and Bourrat, 1998*; *Ishikawa et al., 1999*). Because performing three-dimensional reconstruction from brain slices is difficult, some inconsistencies in nomenclature were observed among references. To overcome this difficulty, we performed a three-dimensional imaging of the nuclei-stained adult medaka telencephalon, which allowed us to analyze the anatomical boundaries in more accuracy. The number in the name of anatomical regions (such as Dm1, Dm2, Dm3) was defined by the order of the emergence in the anterior to posterior axis. Here, the table shows the nomenclature of the anatomical regions of the adult telencephalon and the description of cellular organization in the anatomical regions, which we used to define the names.

| Abbreviation of anatomical region | Name of subregion | Name of anatomical region | Cellular distribution |
|---|---|---|---|
| Dc | | The dorso-central telencephalon | In most teleostean species, cells in Dc have larger size of cell body (Cichlid fish *Burmeister et al., 2009*, sea bass *Cerdá-Reverter et al., 2001*) and are sparsely distributed. According to the medaka brain atlas of *Ishikawa et al., 1999*, Dc is defined in the center of the pallium as well. However, in the other brain atlas of medaka fish (*Anken and Bourrat, 1998*), multiple subregions are separately defined as Dcs. In our atlas, we defined the center region of the pallium that has less dense cell populations as Dc. |
| Dcpm | | The posterior medial nucleus of the dorso-central telencephalon | The aggregates of cells in the posterior medial center of dorsal pallium (Dcpm) were found. |
| Dm | | The medial part of the dorsal pallium | Small cell-body neurons with high density were observed. In the horizontal optic sections, we found several linearly aligned cells on the dorsal surface of the telencephalic hemispheres which correspond to the boundaries of Dm subregions. |
| | Dm1 | | Densely packed with small cells. |
| | Dm2 | | Dm2 is packed with cells more densely than Dm1. |
| | Dm3 | | |
| | Dm4 | | |
| Dl | | The dorsal lateral pallium | In the previous medaka brain atlas, the anterior region of dorsal lateral pallium is simply named as Dl (*Anken and Bourrat, 1998*). But here we named this anterior part of dorso-lateral telencephalon the Dla because the nuclear density is less than Dld, Dlv, and Dlp. We also divided Dl into the dorsal and ventral part since the nuclear density is different between the dorsal and ventral parts. |
| | Dla | The anterior part of the dorso-lateral telencephalon | In the horizontal sections, we found that the density of nucleuses are more sparse in Dla than Dld and Dlv. |
| | Dld | The dorsal part of the middle and posterior part of the dorso-lateral telencephalon | The dorsal part of the middle and posterior part of the dorso-lateral telencephalon (Dld) (*Anken and Bourrat, 1998*) is next to Dla and the cell density is more than that of Dla. |
| | Dlv | The ventral part of the middle and posterior part of the dorso-lateral telencephalon | The boundary between Dld and Dlv is not clear. But the cells are more densely distributed in Dlv than Dld. The ventral part of the middle and posterior part of the dorso-lateral telencephalon (Dlv)(*Anken and Bourrat, 1998*) is highly packed with cells. |
| | Dlp | The posterior regions of the dorso-lateral telencephalon | Higher DAPI signal density in Dlp than Dld and Dlv were detected. There is no clear boundary among the posterior regions of the dorso-lateral telencephalon (Dlp), the posterior of dorsal telencephalon (Dp), Dld, and Dlv. But we observed that Dlp was highly dense with a small nucleus and the cell distribution pattern was different from neighboring regions. |
| Dd | | The dorso-dorsal telencephalon | In the previous report (*Anken and Bourrat, 1998*), Dd is subdivided into two regions, Dd1 and Dd2. In the other report (*Ishikawa et al., 1999*), only Dd is defined which corresponds to Dd2 of *Anken and Bourrat, 1998*. Since the boundary between Dd1 and Dd2 was visible in DAPI staining, we followed the definition of Ralf H Anken, 1998. |
| | Dd1 | | |
| | Dd2 | | Dd2 is clearly demarcated in the telencephalon. As observed with the horizontal sections, Dd2 can be subdivided into two regions, Dd2a (anterior part of Dd2) and Dd2p (posterior part of Dd2). Dd2p is surrounded by cells, and the cell density is relatively higher than Dd2a. |
| Dp | | The posterior part of dorsal telencephalon | The posterior part of dorsal telencephalon (Dp) (*Anken and Bourrat, 1998*) is remarkably denser with the cell nucleus than other posterior anatomical regions. |
| Vd | | The dorsal medial part of the subpallium | Cells in the dorsal medial part of the subpallium are called Vd. We also observed some cell clusters that are located laterally and inside the pallium. Those cell clusters correspond to the Vc region in some references. But according to the cell density and some gene expression, we define those regions also as Vd. |
| Vs | | The supracommissural part of ventral telencephalon | The supracommissural part of ventral telencephalon (Vs) is located ventral to Vd. But there is no clear boundary between Vs and Vd. |

*Table 2 continued on next page*

*Table 2 continued*

| Abbreviation of anatomical region | Name of subregion | Name of anatomical region | Cellular distribution |
|---|---|---|---|
| Vv | | The ventral part of the subpallium | |
| ECL | | The external layer of the olfactory bulb | In the anterior part of the telencephalon, the external (ECL) and internal (ICL) cellular layer of the olfactory bulb and the glomerular layer of the olfactory bulb (GL) are clearly found. |
| ICL | | The internal layer of the olfactory bulb | |

The cell-body distributions in the pallium and subpallium are consistent with the pattern of the neural stem cell (radial glial) (*Figure 1—figure supplement 4*). In the teleost telencephalon, the cell bodies of radial glia locate in the surface of the hemispheres and project inside the telencephalon (*Dirian et al., 2014*). Since neural progenitors migrate along those axons, it is consistent that the cell bodies of the pallial clonally related units are clustered along those axons in a cylindrical way.

## Cell lineages in the pallium tend to project to the same target regions

We next asked whether individual members of cell lineages exhibit similar projection patterns or whether they target various regions. In *Drosophila*, clonal units project to the same target and work as connecting modules (*Ito et al., 2013*), whereas clonally related cells in the mammalian visual cortex tend to connect to each other but do not project to the same targets (*Yu et al., 2012*). To assess this question in teleosts, we traced the axonal projections of the cell lineages (*Figure 2A*). We found that, first, axons and dendrites from the cell lineages in the subpallium intermingled with each other, and they send brain-wide projections (*Figure 2B*).

On the other hand, the pallial cell lineages sent axon bundles mostly to the same target region (*Figure 2*, *Figure 1—figure supplement 3G*). We also found several long connections: from the posterior area of the posterior part of dorsal telencephalon (Dp) to the contralateral hemisphere via the anterior commissure; from the anterior area of Dp and the ventral part of Dlv through Dc to Olfactory bulb. These long projections imply that this transgenic line labels both neural progenitors and young mature neurons at least in some brain regions. Further, we found many connections that project from and into the dorsal pallium (Dd2), which allow us to hypothesize a local pallial network around Dd2 (*Figure 2B*). Based on this observation that

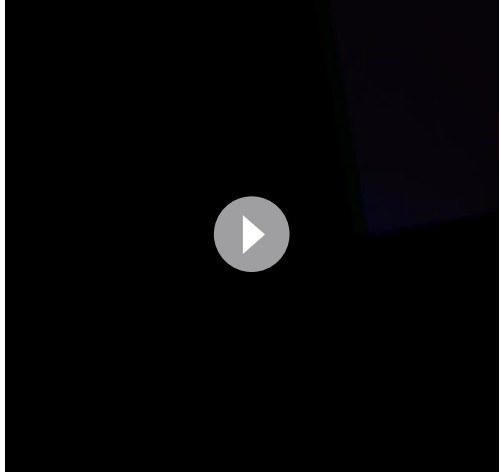

**Video 2.** Pallium. One of the examples of visualized clonally related units in the pallium (raw movie). GFP-positive cells (green) make a cluster and send a projection to the same site. Red signals indicate DsRed and blue signals indicate DAPI staining. The slices show the horizontal view from the dorsal to the ventral direction.

https://elifesciences.org/articles/85093/figures#video2

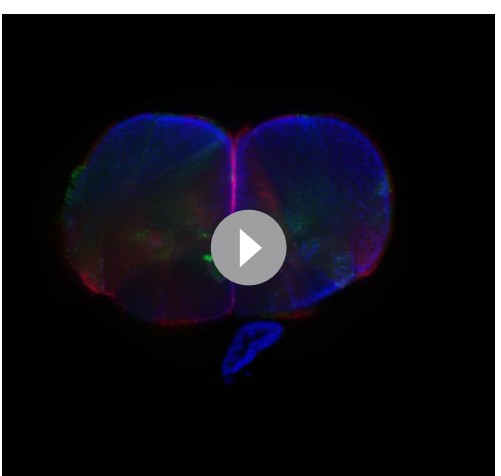

**Video 3.** Subpallium. One of the examples of visualized clonally related units in the subpallium (raw movie). GFP-positive cells (green) and DsRed-positive cells (red) were mixed in the subpallial region. Blue signals indicate DAPI staining. The slices show the horizontal view from the dorsal to the ventral direction.

https://elifesciences.org/articles/85093/figures#video3

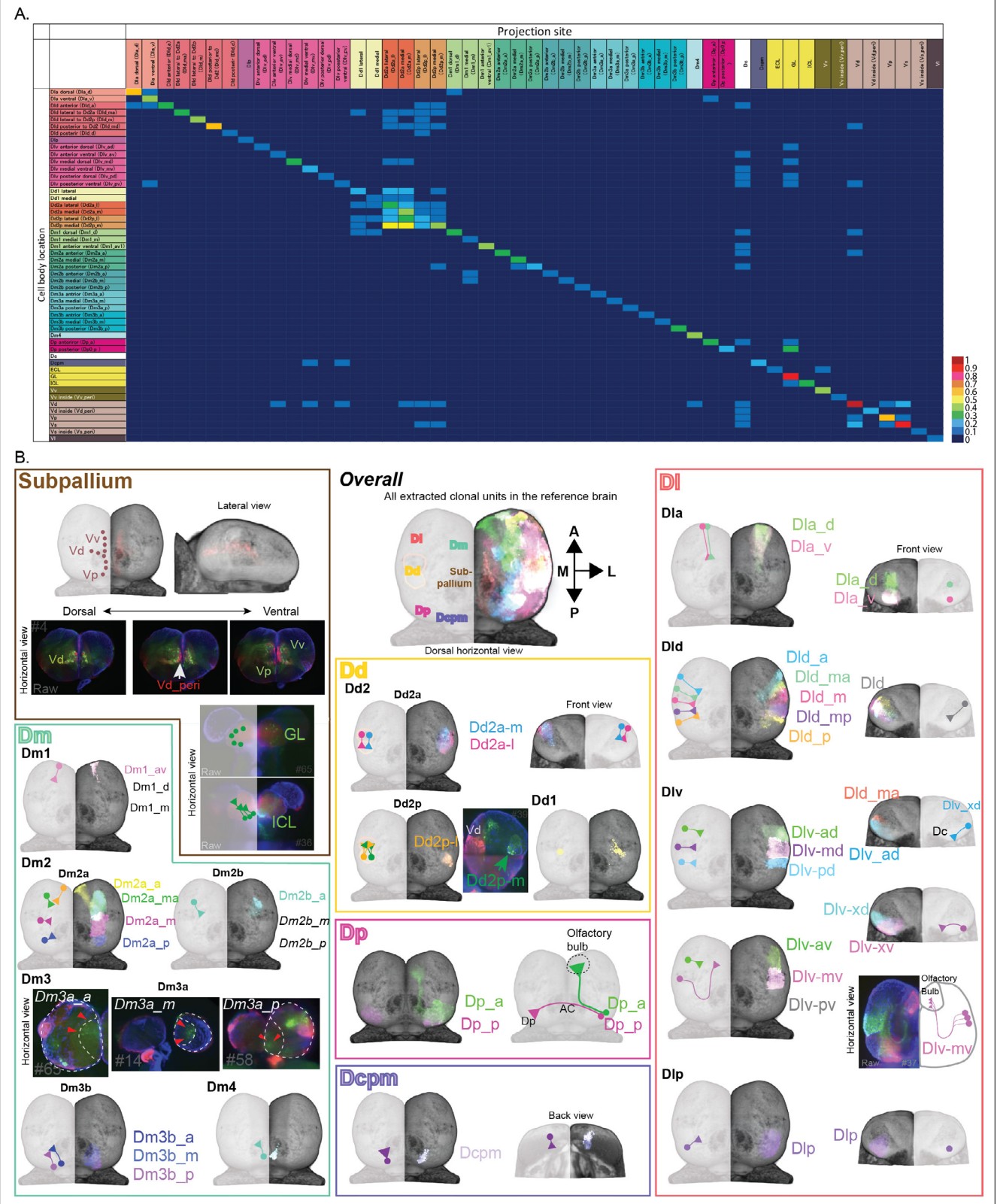

**Figure 2.** Neuronal projections of clonal units in the telencephalon. (**A**) Connectional matrix between clonal units in the telencephalon. The rows indicate the cell body locations and the columns indicate where the projection ends. The number of the projection among 81 fish we analyzed was normalized by the total number of clonal units observed and calculated as the ratio of projection, which is visualized in the color code. (**B**) The structure of clonal units in the telencephalon. We identified the structure of clonal units by systematic analysis, and here we visualized them in different colors

*Figure 2 continued on next page*

Figure 2 continued

that form a single anatomical region. The color of the outline of panels represents the color code of the anatomical regions, but the colors of each clonal unit inside each region are randomly assigned. In each anatomical brain region, the cell soma location and neural projections of clonal units are indicated by circles and line with a triangle terminal for each. For some clonal units where the registration of brains did not work well, we could not visualize the structure of clonal units in the reference brain. In these cases, we showed raw images of those clonal units.

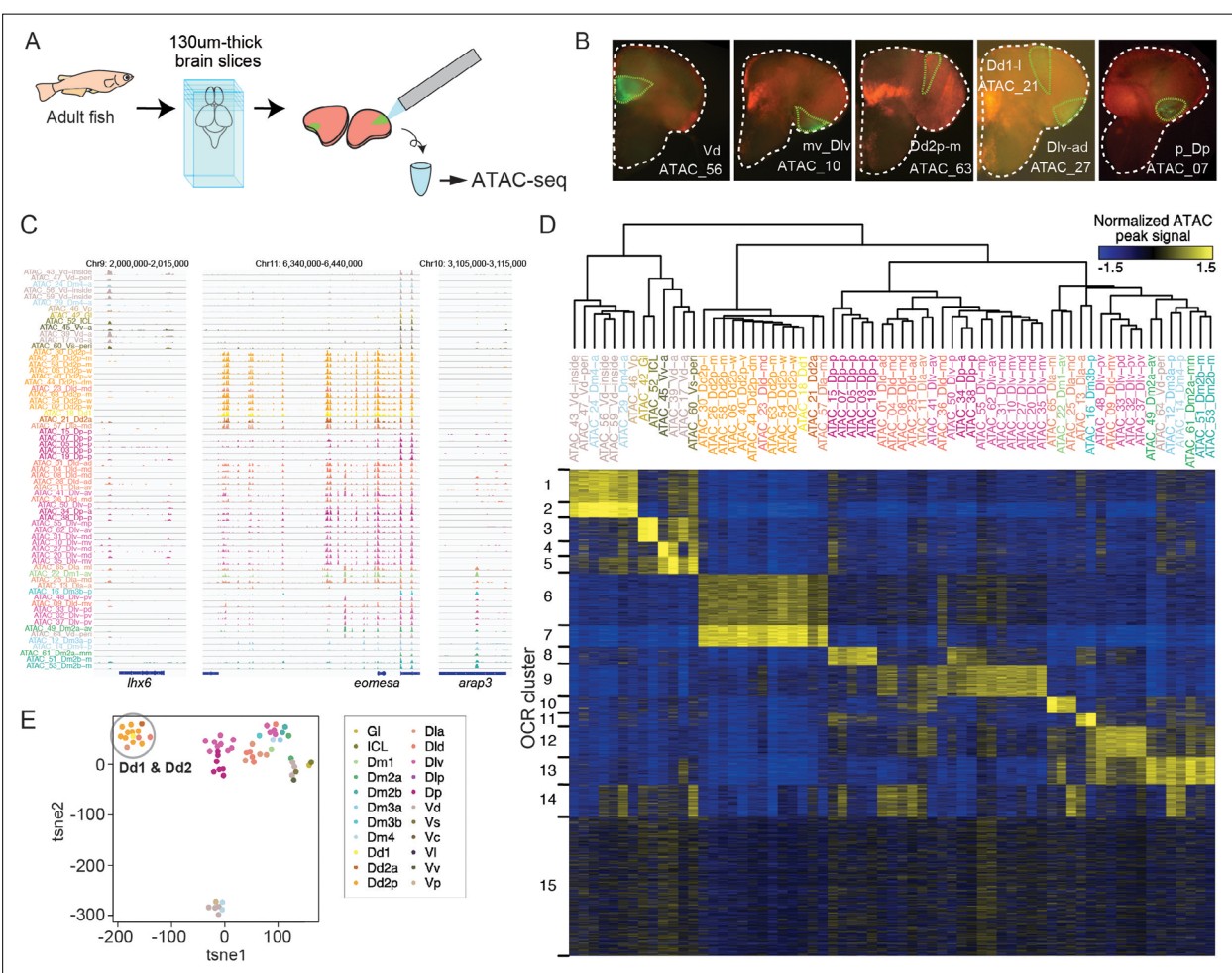

**Figure 3.** ATAC-seq of clonal units in the adult medaka telencephalon. (**A**) Procedure of ATAC-sequencing with extracted clonal units from the adult telencephalon. Cre-loxP recombination was induced at the neurula stage in the transgenic embryos (Tg (HSP-Cre) × Tg (HuC:loxP-DsRed-loxP-GFP)) as previously described. After making 130-um-thick brain slices at the adult stage, GFP-positive cellular subpopulations were dissected and extracted manually. (**B**) Examples of GFP-positive clonal units in the brain slices. White dotted lines indicate the outline of the brain slices. Green lines indicate the dissected GFP-positive clonal units. The ids of clonal-unit ATAC-seq samples are written next to the GFP-positive clonal units in each picture. (**C**) Representative track view of ATAC-seq. Colors of clonal units' names indicate the location in the anatomical regions (*Figure 1A*) of the extracted clonal units. (**D**) Hierarchical clustering and k-means clustering of ATAC-seq peaks in clonal units. In the heatmap, blue indicates closed chromatin regions and yellow indicates open chromatin regions (OCR). Colors of clonal units' names indicate the anatomical regions. (**E**) Dimensionality reduction analysis (tSNE) of ATAC-seq peak patterns of clonal units. Colors of clonal units' names indicate the anatomical regions. A gray circle highlights the datapoints of clonal units of Dd1 and Dd2.

The online version of this article includes the following figure supplement(s) for figure 3:

**Figure supplement 1.** Analysis of ATAC-seq in clonal units.

cell lineages in the pallium tend to project to the same target, we concluded that the labeled cellular subpopulations are clonal units.

## Distinct open chromatin structure in the medaka dorsal pallium (Dd2)

The results enumerated above prompted us to hypothesize that gene expression is uniquely regulated in each clonal unit (*Figure 3*). In order to test this, we performed ATAC-seq on clonal units dissected from sliced brains to examine open chromatin regions (OCRs) that are either specific to or shared across the individually labeled clonal units (*Figure 3A and B*, *Figure 3—figure supplement 1A and B*). We generated ATAC-seq data from 100 sliced brain samples, and 65 samples met the criteria for screening high-quality data (see 'Methods'; *Figure 3C*). To quantitatively test the relationship between the samples, we compared the ATAC-seq peak pattern (see 'Methods'). Hierarchical clustering revealed that clonal units from the same brain regions tend to cluster together (*Figure 3D*). We then classified OCRs into 15 clusters (OCR Cs) by k-means clustering. We first found that OCR clusters specific in the subpallium (OCR C1-5) are distinct from OCR clusters specific in the pallial clonal units (OCR C6-13), which is reflected by their distinct gene expression from the early developmental stage (*Mueller and Wullimann, 2009*).

Within the pallium, clonal units in the different subregions show unique patterns, where the medial and posterior pallium contained OCR C13 and OCR C8, respectively, and the ventral-lateral pallium (Dlv), the medial (Dlv-m), and posterior (Dlv-p) showed a variety of different OCRs (*Figure 3D*).

Remarkably, the clonal units in the dorsal pallium (Dd2) were found to have distinct chromatin structure compared to all other clonal units in the telencephalon (*Figure 3D*); OCR C6 and 7 were specifically open in the genome of Dd2 samples, and other OCR clusters that are open in other pallial regions (e.g. C8-14) tended to be closed in Dd2. Dimensionality reduction methods, such as principal component analysis (PCA), tSNE, and Uniform Manifold Approximation and Projection (UMAP) analysis, confirmed that, apart from subpallial clonal units, clonal units in Dd2 were distant from other pallial cell lineages (*Figure 3E*, *Figure 3—figure supplement 1C*). We examined the distribution of ATAC-seq peaks in the genome, and Dd2-specific ATAC-seq peaks, located mainly in the intron and intergenic regions (*Figure 3—figure supplement 1D*), suggesting that the genomic regions that exhibit these peaks function as Dd2-specific enhancers. In summary, this ATAC-seq analysis suggests that the chromatin structures differ, depending on brain regions, and that it singles out the Dd2 region as an area of particularly distinct regulation.

## Enhanced transcriptional regulation on synaptic genes in Dd2

Next, to uncover which genes are differentially regulated in Dd2, we examined Gene Ontology (GO) term enrichment for genes targeted by each OCR cluster (*Figure 4A*). First, we found that the terms related to axon guidance and neurogenesis were enriched in various OCR clusters (*Figure 4A*, top; OCR C1, 3–7, 9, 12, and 14), which suggests that genes in the axon guidance pathways and neurogenesis are differentially regulated in different clonal units. Also, the GO terms related to early development were enriched in OCR cluster 14, and many metabolic and biosynthesis processes were enriched in OCR cluster 15. Also, one of the subpallial OCR clusters (OCR C5) was enriched with neuropeptide signaling, neural differentiation, and vascular processes (*Figure 4—figure supplement 1*).

Intriguingly, one of the Dd2-specific OCR clusters, OCR C6, was enriched with GO terms related to synaptic signaling (*Figure 4A*, bottom). As OCRs reflect the transcriptional regulation in both enhancing and suppressing directions, we analyzed the actual expression of synaptic signaling genes in Dd2 by RNA-seq on Dd2, the pallium regions excluding Dd2 (D), and the subpallium (V) (*Figure 4—figure supplement 2A and B*). First, GO term enrichment analysis shows that synaptic signaling-related terms are significantly enriched in genes preferentially expressed in Dd2 (i.e. expressed significantly higher in Dd2 than D) (*Figure 4B*). Among 82 synaptic genes preferentially expressed in Dd2, 68% of them were targeted by OCR C6 or C7, which we call 'actively upregulated synaptic genes'.' On the other hand, 70 synaptic genes were significantly lowly expressed in Dd2 compared to D, and 40% of them were the targets of OCR C6, which we call 'actively downregulated synaptic genes' (*Figure 4—figure supplement 2C*).

Since the transcription of synaptic genes is actively regulated in Dd2, we checked whether synaptic density is higher in the dorsal pallium (*Figure 4C*, *Figure 4—figure supplement 2D*). We observed the high expression of synaptic markers in both larval and adult telencephalon in medaka; presynaptic

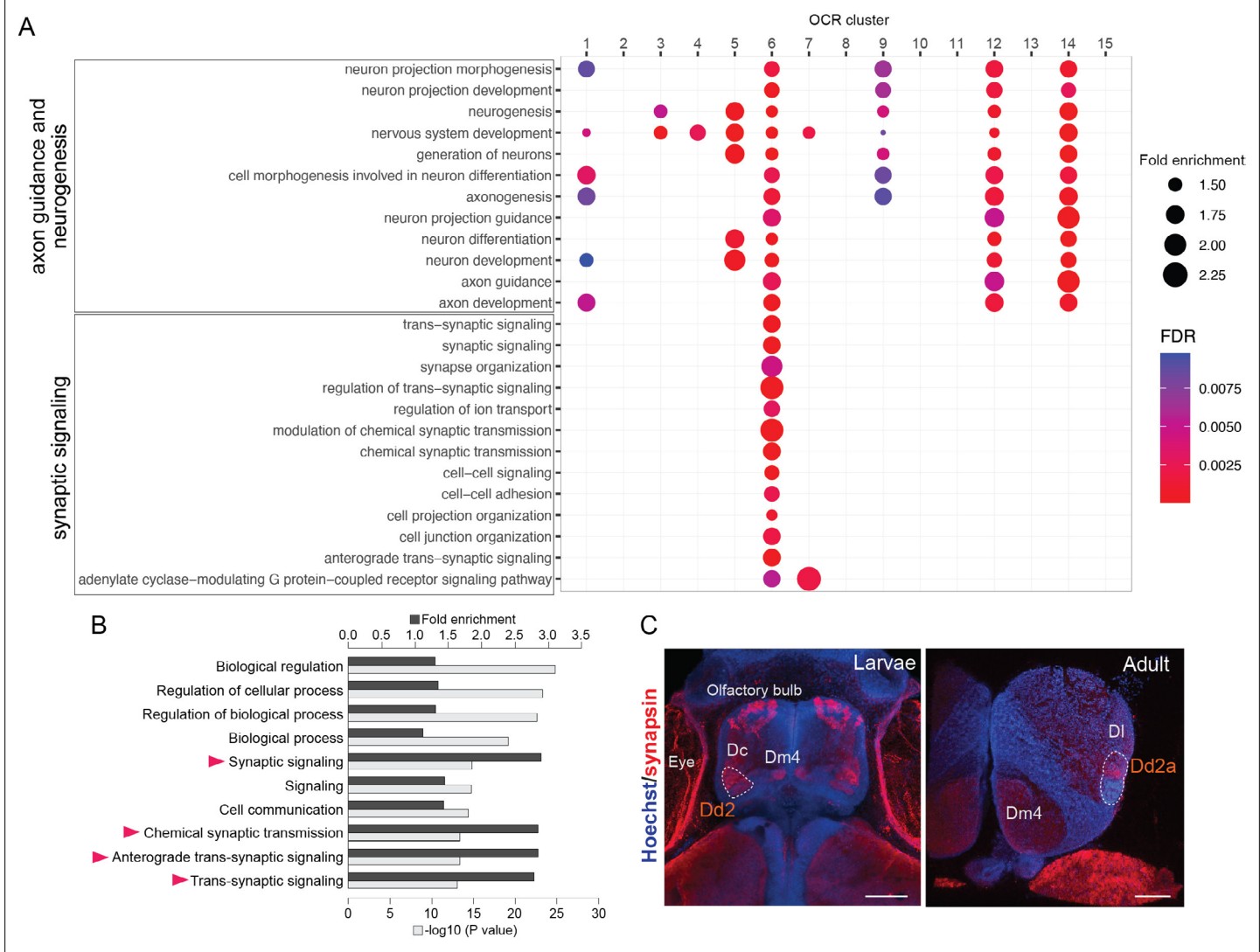

**Figure 4.** Analyses of Dd2-specific open chromatin region (OCR) clusters reveal specialized synaptic gene transcription. (**A**) Gene Ontology (GO) analysis of OCR clusters. GO terms related to axon guidance and neurogenesis (top) and synaptic signaling (bottom) are shown (see also *Figure 4— figure supplement 1* for other enriched GO terms). Size and color of circles indicate fold enrichment and false discovery rate (FDR), respectively. Circle was plotted only if FDR is lower than 0.01. GO terms of axon guidance and neurogenesis are enriched with pallial clusters (OCR C9 and 12), Dd2 cluster (OCR C6), subpallial cluster (OCR C1, 3, 4, 5), and a common cluster (OCR C14). On the other hand, synaptic genes are enriched in OCR C6. (**B**) GO analysis of the genes preferentially expressed in Dd2. Top 10 significantly enriched terms are shown. Black bars indicate the fold enrichment and gray bars show the log p-value. GO terms related to synaptic signaling are highlighted with triangles. (**C**) Anti-synapsin immunohistochemistry on the medaka larvae and adult telencephalon. Blue: Hoechst; red: anti-synapsin signals. In larvae, strong signals were detected in the olfactory bulbs, Dc, Dm4, and Dd2. In the adult telencephalon, the signals were broadly detected and strongly detected in Dd2a. Scale bar in larvae: 50um, scale bar in adult: 200um.

The online version of this article includes the following figure supplement(s) for figure 4:

**Figure supplement 1.** Gene Ontology (GO) term enrichment analysis on open chromatin region (OCR) clusters.

**Figure supplement 2.** RNA-seq and ATAC-seq analysis on Dd2.

marker synapsin in Dd2a, and postsynaptic marker PSD95 in both Dd2a and Dd2p (*Figure 4—figure supplement 2D*). Also, to see whether this synapse-enriched area in the dorsal pallium is unique in medaka, we also examined the expression of synaptic marker genes in zebrafish larvae, and PSD95- or synapsin signals were detected in Dc but not in the dorsal pallium in zebrafish.

In the end, to characterize the synaptic property in the dorsal pallium, we assessed the 'actively regulated synaptic genes' in Dd2 (*Figure 5A and B*, *Figure 5—figure supplement 1*). First, RNA-seq data showed differential expression of several synaptic genes in many types of neurons (modulatory,

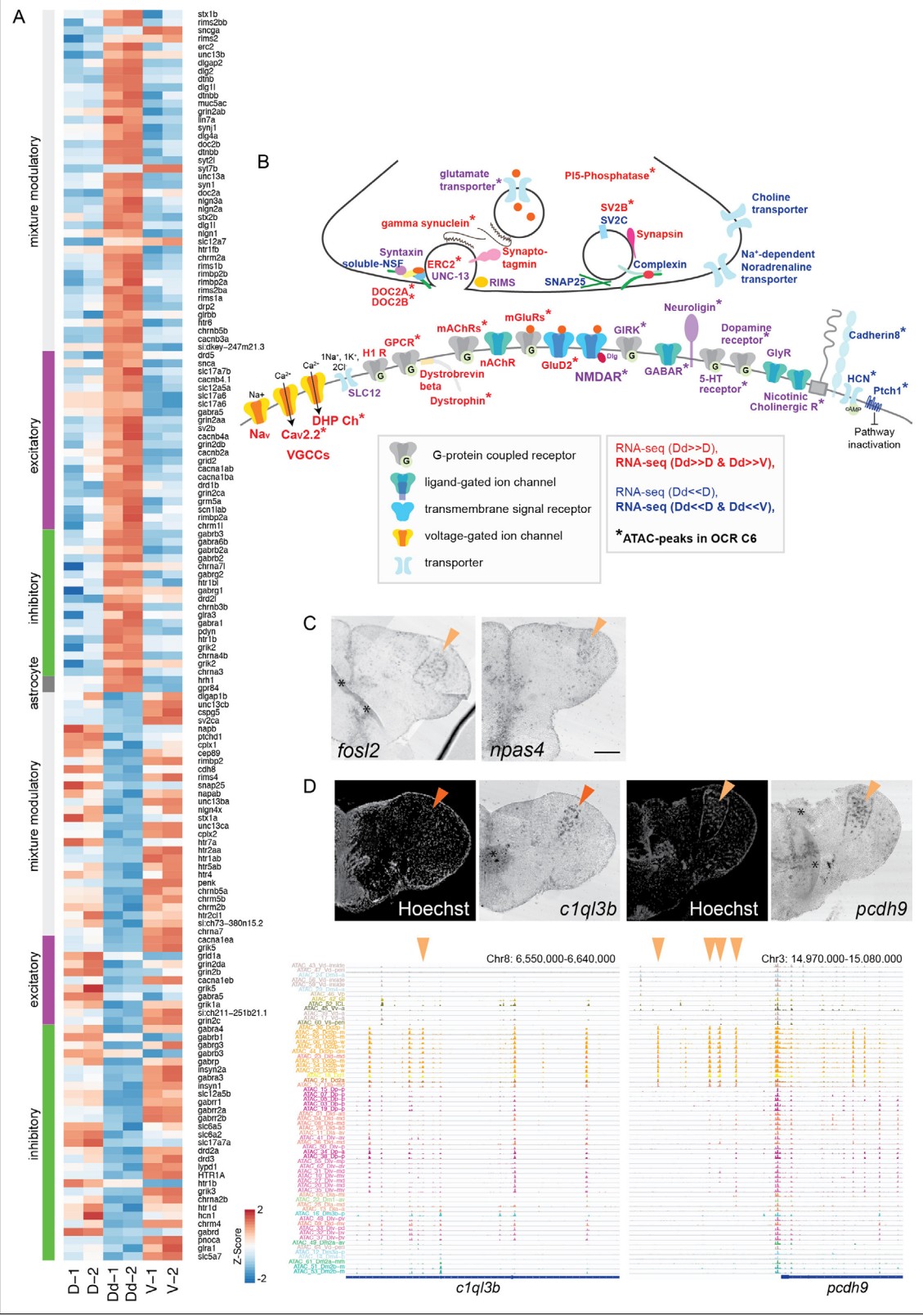

**Figure 5.** Analyses of Dd2-specific open chromatin region (OCR) clusters reveal specialized synaptic architecture. (**A**) Differential expression of synaptic signaling genes in the Dd2 compared to other pallial regions and the subpallium. Expression levels from RNA-seq data are shown for synaptic signaling genes that were differentially expressed in Dd2. Synaptic genes are classified into excitatory, inhibitory, modulatory, and astrocyte categories. (**B**) Schematic summary of the genes actively regulated or preferentially expressing genes in Dd2 regions. Gene names in red indicate all subunits in

*Figure 5 continued on next page*

*Figure 5 continued*

the gene expressed significantly higher in Dd2 than in other pallial regions, while gene names in blue indicate the expression of all subunits in Dd2 was significantly lower in Dd2. Gene names in purple indicate that the gene included both higher and lower expressing subunits. * indicates the genes that are actively regulated by the Dd2-specific OCR cluster (C6). More detailed information about the differentially regulated subunits of genes is described in *Figure 5—figure supplement 1*. GABAR: GABAergic receptor; GIRK: glutamate receptor ionotropic kainate; GluD2: delta glutamate receptor2; GlyR: glycine receptor; H1R: histamine receptor; mAChRs: muscarinic acetylcholine receptors; nAChR: nicotine acetylcholine receptor; SLC12: kidney-specific sodium-potassium-chloride cotransporter; CaV2.2: N-type voltage-gated calcium channels; DHP Ch: L-type voltage-gated calcium channels; VGCCs: voltage-gated calcium channels; Nav: voltage-gated sodium ion channels; PI5-phosphatase: phosphoinositide 5-phosphatase. (**C**) Expression of immediate early genes (fosl2 and npas4) in the Dd2p region (orange triangles). Scale bar: 200um. (**D**) Expression of synaptic genes specifically expressed in Dd2 visualized by in situ hybridization (top). c1ql3b was specifically expressed in Dd2a (top left), and protocadherin 9 was detected in Dd2p (top right). Orange triangle indicates the gene expression in Dd2. Track view of ATAC-seq peaks in clonal units (bottom). Orange triangles indicate the Dd2-specific ATAC-seq peaks.

The online version of this article includes the following figure supplement(s) for figure 5:

**Figure supplement 1.** Characterization of synaptic genes expressed and regulated in Dd2.

excitatory, and inhibitory neurons) (*Figure 5A*). Then combining RNA-seq with ATAC-seq data, we found that various synaptic genes are targeted by OCR C6 (*Figure 5B*): subunits of the glutamate receptors, transporters, and modulatory neurotransmitter receptors (5-HT receptors, cholinergic receptors, and dopamine receptors) were differentially regulated in both positive and negative directions. On the other hand, a variety of voltage-dependent calcium channels expressed significantly higher in Dd2 than other pallial regions, whereas many inhibitory synaptic genes were expressed significantly lower, such as *HCN* and *Ptch1* (*Figure 5A and B*, *Figure 5—figure supplement 1*). These results implied that neurons in Dd2 maintain persistent activity, which is consistent with the expression of some immediate early genes in Dd2 (*Figure 5C*). Lastly, among the synaptic genes specifically expressed in Dd2 and accompanied by Dd2-specific ATAC-seq peaks, we examined the expression of a few genes by in situ hybridization (ISH) (*Figure 5D*); synaptic regulator, *c1ql3b*, in Dd2a; a protocadherin gene, *pcdh9*, in Dd2p, and confirmed region-specific expressions in Dd2.

## Transcriptional regulators of Dd2-specific OCR clusters

We focused our attention on the mechanism that regulates Dd2-specific gene transcriptions (*Figure 6*). First, we used ATAC-seq data of clonal units to identify candidates of regulators by searching the known TF binding motifs enriched in the OCR clusters (*Figure 6A*). In the Dd2-specific OCR clusters (C6 and 7), we found that binding motifs of beta helix-loop-helix (bHLH), zinc finger (Zf), and T-box families were significantly enriched, suggesting that multiple TFs regulate gene expression in Dd2 neurons. We also performed de novo motif searching in Dd2-specific OCR clusters and found that the result was consistent with the known-motif search (*Figure 6B*).

Next, we assessed the actual expression of the candidate regulators of Dd2. In the RNA-seq data, we found that TFs whose binding motifs that were enriched in Dd2-specific OCR clusters were preferentially detected in the Dd2 sample (*Figure 6C*). Then, we examined the expression of candidate TFs by ISH (*Figure 6D*, *Figure 6—figure supplement 1A*). Though we did not find a transcription factor that is only expressed in Dd2 regions, we found several candidate TFs expressed strongly in Dd2; *tshz-1* (Zf family) expressed strongly in Dd2 and weakly in Dl; several other Zf families, such as *egr-1*, *npas4*, and *znf827*, expressed in Dd2 and other part of the telencephalon; *eomesa* (T-box family) expressed in Dd, Dl, and Dp. *TSHZ* is a member of the C2H2-type zinc-finger protein family. Although its specific binding motif is not yet known, it is possible that *TSHZ* shares similarities in binding motif with other C2H2 zinc-finger proteins. Interestingly, the motif of *Egr*, a C2H2 zinc-finger protein, was found to be highly enriched in Dd2-specific OCR clusters (C6 and 7; *Figure 6B*), which suggest a potential involvement of *tshz-1* in the differentiation of the Dd2 region.

Finally, we questioned whether medaka Dd2 is homologous to any regions in the mammalian pallium. We compared putative transcriptional regulators between medaka clonal units and human brain regions from *Fullard et al., 2018* by investigating the TF motif enrichment in the OCRs (*Figure 6E*). We compared the significantly enriched TF motifs between medaka OCR clusters and human differential OCRs (*Fullard et al., 2018*) by counting the number of overlaps. We found that our medaka OCR clusters shared only a few motifs with the human medial dorsal thalamus (MDT), which is consistent with the fact that the thalamus does not belong to the telencephalon (*Figure 6—figure*

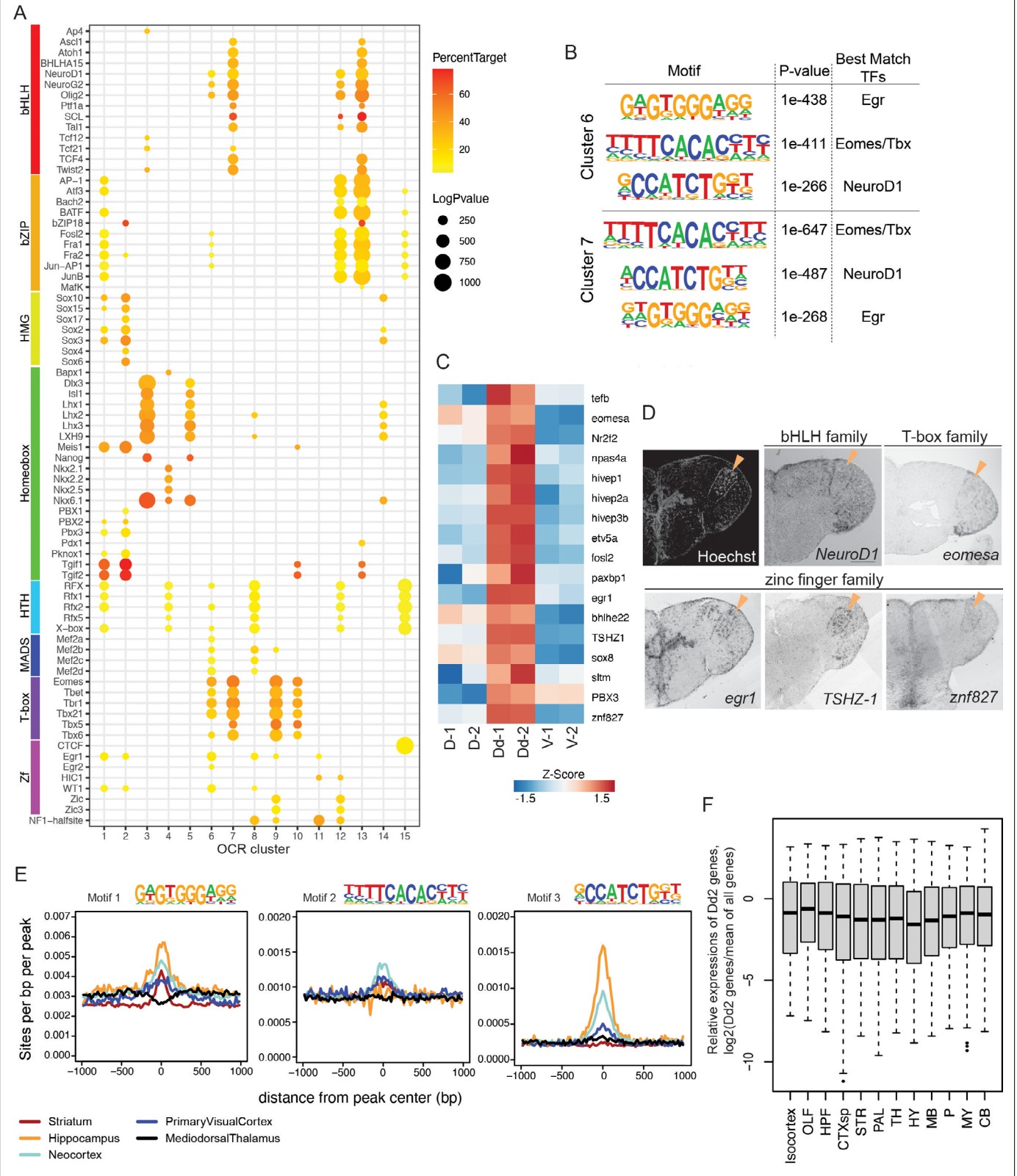

**Figure 6.** Transcriptional regulation in Dd2. (**A**) Enrichment of known transcription factor binding motifs in open chromatin region (OCR) clusters are shown as dot plots. Size and color of circles indicate -log₁₀(p-value) and percent target, respectively. Circle was plotted only if -log₁₀(p-value) is higher than 100. (**B**) Result of de novo motif finding analysis. Top 3 of the enriched motifs in OCR C6 and 7 and candidate transcription factors that have similar binding motifs are shown. (**C**) Expression of candidate transcription factors in Dd, the pallium except Dd (D), and the subpallium (V) analyzed by

*Figure 6 continued on next page*

*Figure 6 continued*

RNA-sequencing (n = 2 replicate). (**D**) Expression pattern of the candidate transcription factors in the telencephalon visualized by in situ hybridization (ISH). Orange triangles indicate the position of Dd2p. (**E**) Medaka OCR C6 motif enrichment in human brain region-specific OCRs. Existence of each medaka C6 motif was examined around human brain ATAC-seq peaks, and the motif density was calculated. HC: hippocampus; MDT: medial dorsal thalamus; NCX: neocortex; PVC: primary visual cortex; ST: striatum. Darker blue shades indicate higher correlation. (**F**) Relative expression levels of Dd2-preferentially expressed genes (809 genes) in the mouse brain. We examined the expression of Dd2-preferentially expressed genes across 12 regions in the mouse brain (isocortex. OLF: olfactory areas; HPF: hippocampal formation; CTXsp: cortical subplate; STR: striatum; PAL: pallidum; TH: thalamus; HY: hypothalamus; MB: midbrain; P: pons; MY: medulla; CB: cerebellum). The gene expression levels were from the ISH data of Allen Brain Atlas (https://mouse.brain-map.org/).

The online version of this article includes the following figure supplement(s) for figure 6:

**Figure supplement 1.** Expression of candidate transcription factors in the adult medaka telencephalon.

*supplement 1B*). Also, we confirmed that the subpallium-specific medaka OCR cluster (C1) shared a relatively larger number of motifs with the human striatum (ST), a part of subpallium in mammals. We found that the medaka Dd2-specific OCR clusters (C6 and C7) share most motifs with the human hippocampus (HC), neocortex (NCX), and ST. However, this analysis might suffer from bias when there are many similar motifs from the same TF family. Thus, we examined the enrichment of Dd2-specific OCR motifs in human brain OCRs. As shown in *Figure 6B*, medaka OCR C6 and C7 were both enriched with three TF motifs that best match to Egr, Eomes/Tbx, and NeuroD1 motifs. We therefore examined the distribution of the motifs 1–3 of OCR C6 in human ATAC-seq peaks. We found that motifs 1 and 3 were enriched in HC and NCX OCRs, and motif 2 was enriched in NCX the most (*Figure 6E*). Additionally, we examined the expression levels of genes that are preferentially expressed in medaka's Dd2 (809 genes) across brain regions in mouse using ISH data from the Allen Mouse Brain Atlas (*Lein et al., 2007*). However, there was no observed gene expression biased toward any specific brain region. Further comparative analysis in synaptic structure, circuit structure, and neural function in the dorsal pallium across vertebrates is indispensable to discuss the homology and explain these shared transcriptional regulatory elements.

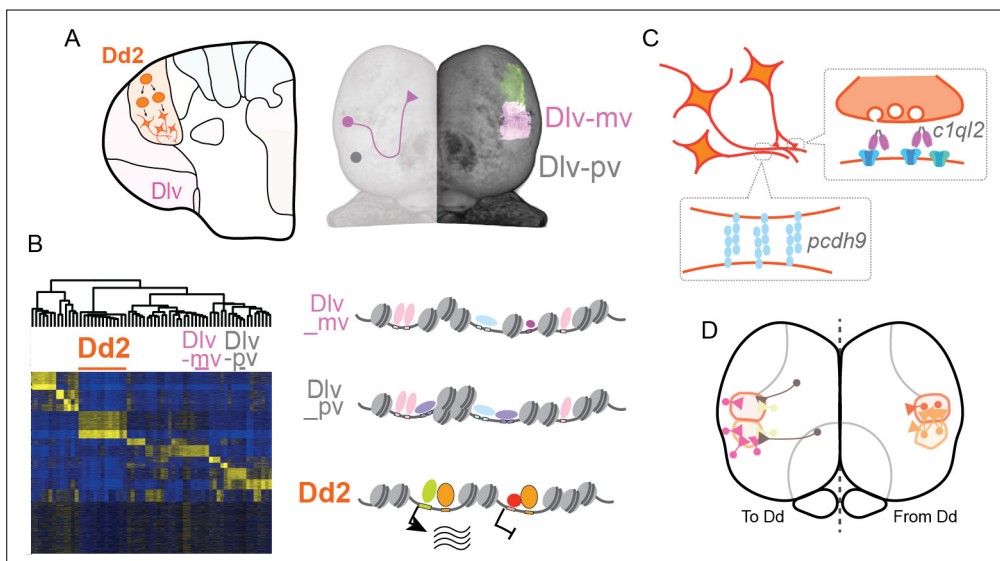

**Figure 7.** Hypothesized molecular model of specialized synaptic architecture in the dorsal pallium in medaka. (**A**) Anatomical regions in the pallium is formed by clonal units mutually exclusively. (**B**) Clonal ATAC-seq shows unique open-chromatin patterns in some clonal units (left). Transcription is likely regulated by the combination of multiple TFs that construct a distinct open chromatin structure (right). (**C**) This specific open chromatin region (OCR) contributes to the transcriptional regulation of a specialized set of synaptic genes in Dd2, which are suggested to generate specific axonal projections and synaptic architecture. (**D**) We assume that Dd2 is important in the telencephalic neural networks in medaka. Circles indicate cell bodies and triangles indicate projection terminals. The color code indicates the position of anatomical regions identical to the color code in *Figure 1*.

## Discussion

Here, we found that the pallial anatomical regions are formed by clonal units mutually exclusively, and in particular, the dorsal pallium (Dd2) in medaka is distinct from other pallial regions in terms of epigenetic regulation, especially in synaptic signaling (*Figure 7*).

### Spatial boundaries in animal bodies

In animal bodies, various boundaries are formed by clonally related cells. In the *Drosophila* brain, clones do not cross segments, suggesting that clones are projection units (*Ito et al., 2013*). In the zebrafish heart (*Gupta and Poss, 2012*), as fish grow from juveniles to adults, the whole heart is formed by cells derived from a small number (~8) of cardiomyocytes, which define the sections through clonal dominance. In the mammalian gastrointestinal tract (*San Roman and Shivdasani, 2011*), several organs, such as esophagus and glandular stomach, are formed by patches of clonal units that exhibit sharp boundaries and serve distinct digestive functions. Different expression patterns of *Hox*-cluster genes contribute to the boundary maintenance. In the medaka retina (*Centanin et al., 2014*), retinal stem cells possess multipotency and generate all neuron types, and clonally related cells form the cylindrical structure (*Centanin et al., 2011*).

We speculate that clonal boundaries in the medaka pallium have been established with the similar mechanism as in medaka's retinal boundaries. In the adult teleost pallium, the radial glia cells on the surface of the telencephalic hemisphere, which project toward inside of the hemisphere, function as neural stem cells, and generate neural progenitors throughout life (*Figure 1—figure supplement 4*). This cellular arrangement might contribute to the compartmentalized structure.

As shown in mammals, the epigenetic landscape can be inherited from apical progenitors, which has a multipotency, to the late neural progenitors during development (*Di Bella et al., 2021*). Since the teleost exhibit post-hatch neurogenesis in the entire life, we think that the common epigenetic landscape is inherited in each clonal unit in the adult medaka telencephalon. As a result, we make the assumption that function and characteristic of each clonal unit are defined already in progenitors by specific regulators (e.g. TFs), and those progenitors continuously produce neurons that possess the same property to function in a coordinated manner.

### Pallial diversity in teleost fish

The pallium in teleosts has been mysterious since different species exhibit different numbers or cellular densities of compartmentalized anatomical subregions within the pallium. On the other hand, the subpallial regions are relatively conserved among teleost species in terms of anatomical subregions and gene expressions (*Moreno et al., 2009*; *Ganz et al., 2012*). Our structural analysis in the adult medaka telencephalon revealed that the clonal architecture between the pallium and subpallium differs in the distribution of cells in clonal units: clonal units in the subpallium intertwine with each other, whereas the pallium is formed by the compartmentalized clonal units, giving rise to a modular structure. Modular structure is frequently seen in animal body, including brain; central complex in insect (*Turner-Evans and Jayaraman, 2016*), cerebellum in vertebrates (*Apps and Garwicz, 2005*). The modularity of cell populations or organs is generally thought to contribute to evolutionary flexibility; one module can acquire a new phenotype without impacting the others (*Bolker, 2000*; *Kuratani, 2009*; *Wagner et al., 2007*). We assume that the modular nature of the clonal units in the pallium plays a key role in the diversity across teleost.

How did the dorsal pallium evolve in teleosts? One hypothesis is that the teleostean dorsal pallium has been specialized from the teleostean lateral pallium (Dl) (*Yamamoto et al., 2007*), whereas an alternative hypothesis proposes that the teleostean dorsal pallium evolved independently (*Striedter and Northcutt, 2022*). Our RNA-seq data identified a couple of TFs strongly expressed in Dd2 in medaka fish. In zebrafish, similar TFs, such as *tshz-1, eomesa*, and *egr-1*, are expressed in Dl (*Diotel et al., 2015*), suggesting that medaka Dd2 shares a part of regulatory mechanism with zebrafish Dl. In addition, our PCA on ATAC-seq data with clonal units showed that open chromatin patterns in medaka Dd are relatively close to that in Dl. Taken together with the modular nature of clonal units in the pallium, our analyses favor the first scenario that Dd has been differentiated from the neighboring anatomical region, Dl.

Since medaka belong to the Beloniformes that diverged later than the Cypriniformes, which includes zebrafish, it might be possible that the last common ancestors of bony fish possess the

dorsal pallium when they diverged, and the dorsal pallium disappeared later or it is not yet identified in Cypriniformes. Further analysis in other fish species will be required to explain the evolutionary trajectory of the dorsal pallium.

## Possible neural computations and functions of medaka Dd2

Numerous computations happen in synapses, and synaptic architecture is defined by the expression of a subset of specific molecules. Some neurons in mammals and songbirds exhibit synapses that are characterized by particularly strong plasticity and modulation, where complicated information processing takes place or memories are stored (*Okamoto et al., 2004*; *Champoux et al., 2021*). In this article, we found that many synaptic genes are differentially regulated in Dd2 and they span the range from excitatory, inhibitory, and modulatory synaptic genes. We also identified a synaptic organizer, *C1ql3*, which expressed specifically in the anterior Dd2 (Dd2a), and also expresses in the mammalian hippocampus where it is known to organize extracellular postsynaptic space and bind to GPCRs in a calcium-dependent way (*Matsuda et al., 2016*; *Ressl et al., 2015*). Taken together, our results suggest that the transcription of synaptic genes in the dorsal pallium is selectively regulated and might assist information processing and activity-dependent plasticity in a manner different from other pallial regions.

What role does the dorsal pallium (Dd2) in medaka play within the telencephalic network? Our findings suggest that Dd2 might function as either a network hub where the flow of information is gated through synaptic modulation (*Gal et al., 2017*), or, alternatively, it might serve as a working memory module that can transiently store information by persistent activity. Several lines of evidence support this notion. First, a couple of immediate early genes are expressed in Dd2. Second, from structural analysis, we observed axonal projections from many surrounding regions into Dd2 as well as dense and, usually unilateral, connectivity between Dd2 subregions (from Dd2p to Dd2a). Third, cadherin molecules are known to play an important role in regionally targeted axonal projection (*Sun et al., 2021*), and we found specific expression of *pcdh9* in Dd2p (*Kim et al., 2011*). Here, it should be noted that our axonal projection analysis might have missed the projections from old mature neurons because the HuC promoter labels preferentially neural progenitors and young neurons, and the labels disappear at later stages (*Okuyama et al., 2013*). Brain-wide exhaustive connectome analysis by electron microscopy (*Odstrcil et al., 2022*) or neurophysiological analysis will be necessary for further study (*Bartoszek et al., 2021*).

At present, few studies have been done to uncover the behavioral function of the dorsal pallium in teleosts: in electric fish, projection analysis suggests that the dorsal pallium possesses hippocampal-like circuitry (*Elliott et al., 2017*). In several cichlid fish species, the dorsal pallium gets activated specifically in the context of social memory tasks (*Rodriguez-Santiago et al., 2021*). Further, some teleostean species that show social behaviors requiring memory exhibit distinct cytoarchitecture in their dorsal pallium. These include cichlid fish that show social dominance (*Rodriguez-Santiago et al., 2020*), and, interestingly, medaka fish show mating behavior involving social memory (*Okuyama et al., 2014*; *Wang and Takeuchi, 2017*; *Yokoi et al., 2020*). It should be the work of the future to study whether Dd2 in medaka functions in the context of social behavior using memory.

## Mammalian brain regions homologous to the teleost dorsal pallium

The homology of the anatomical region in the telencephalon across vertebrates has been a subject of intense debate. Especially the correspondence of the pallial regions between teleosts and amniotes has been controversial since the growth of neural tubes in early development proceeds in the opposite direction, such as evagination in amniotes and eversion in teleost (*Striedter and Northcutt, 2019*; *Striedter, 2005*).

Recent studies in zebrafish provide more evidence that the medial pallium (Dm) is homologous to the amygdala and the lateral pallium (Dl) is homologous to the hippocampus. Dm gets activated with fear conditioning and anxiety-related experiments (*Lau et al., 2011*; *Lal et al., 2018*; *Reichmann et al., 2020*; *Baker and Wong, 2021*) and Dl gets activated in learning paradigm (*Dempsey et al., 2022*). Though zebrafish do not have clear boundaries in the dorsal part of the pallium, there are high activity in the central part of the pallium (Dc) in many behavioral paradigm, such as prey capture (*Oldfield et al., 2020*), multimodal avoidance behavior (*Randlett et al., 2015*), and even locomotive behavior (*Dunn et al., 2016*), which could correspond to the ongoing neural activity in the mammalian

cortex (*Hiraki-Kajiyama et al., 2019*). Also based on gene expression, some studies imply that Dc is homologous to the mammalian dorsal pallium (*Porter and Mueller, 2020*), and we detected high synaptic density in Dc (*Figure 4—figure supplement 2*). On the other hand, medaka have clear boundaries in the dorsal pallium (Dd2). From the topological point of view, it is possible that Dd2 in medaka may be related to either the hippocampus or neocortex in mammals.

In our study, we found a specific transcriptional regulation in synapses in Dd2 in medaka, which could be a similar feature of the mammalian hippocampus and neocortex; some studies in mammals suggest that synaptic properties are regulated heavily in the hippocampus and neocortex, where density and size (*Holler et al., 2021*) of synapses are controlled, presumably in the context of synaptic plasticity (*Neves et al., 2008*). Also, our motif analysis provides insight that medaka Dd2-specific OCRs share the binding motifs of transcriptional regulators with the human hippocampus and neocortex. On the other hand, we did not find a significant correlation in the overall gene expressions between medaka Dd2 and any of the mammalian brain regions. However, we believe further investigation on the synaptic regulations in specific circuits in the dorsal pallium will allow us to uncover whether the primitive region of either hippocampus, neocortex, or the mixture of two is conserved in the dorsal pallium in the teleost or not.

## Materials and methods

### Fish husbandry

Medaka fish are kept at 14 hr/10 hr on light at 28°C. The transgenic line (Tg) (HuC:loxp-DsRed-loxp-GFP) and Tg (HSP:Cre/Crystallin-CFP) were generated as previously reported (*Okuyama et al., 2013*). Both males and females were used in the experiments. We used more than 2-month-old fish as adult fish.

### Clonal units visualization

To induce Cre/loxP recombination, embryos of the double Tg line, being crossed from female Tg (HuC:loxp-DsRed-loxp-GFP) and male Tg (HSP:Cre) line. When the embryos developed to the neurula stage, they were mildly heated at 38°C for 15 min. Heat shock was applied using a thermal cycler for polymerase chain reaction tubes containing two eggs each. After the mild heat shock treatment, the embryos were maintained at 26°C until they hatched. Then the fish were kept in the fish tank and raised till the adult stage.

### Whole-brain clearing and imaging

Adult fish brains were dissected by an established procedure (*San Roman and Shivdasani, 2011*). After the dissection, the brains were fixed in 4% paraformaldehyde/phosphate-buffered saline (PBS) overnight. Then the brains were washed twice with PBST (0.5% Triton-X100 in PBS) and immersed into ScaleA2 solution for about 3 hr on ice (*Hama et al., 2011*). When the brains were confirmed to be transparent, they were transferred into a new ScaleA2 solution with DAPI (0.5 ul/2 ml) overnight at 4°C. The next day, the brains were transferred into the new ScaleA2 solution and washed with PBST. To prepare samples for signal detection, the brains were embedded in 0.75% agarose gel with ScalA2-Triton solution and fixed on glass coverslips. Fluorescent signal detection was performed using a hand-made light sheet microscopy digital light-sheet microscope (DSLM) (*Ichikawa et al., 2013*) and a commercial one (Zeiss, Lightsheet Z.1).

In order to define the anatomical regions, we used DAPI signals as landmarks. For whole-brain imaging, we dissected adult medaka brains, fixed with 4% paraformaldehyde/PBS, stained with DAPI in ScaleA4 (0.5/2000) for 2 d, and soaked with ScaleA4 for several days.

All original data used for the clonal analysis is accessible in Dryad Digital Repository (doi:10.5061/dryad.ttdz08m2s).

### Cell density assessment

Ilastik (*Fiaschi, 2012*) was used to examine how many cells are labeled in the Tg (HuC:loxp-DsRed-loxp-GFP). We dissected and fixed the adult brain in 4% PFA/PBS. Then after washing, we performed clearing with Scale A2 (*Hama et al., 2011*) and stained the brain with DAPI. After washing, fluorescent signal detection was performed using a hand-made light sheet microscopy (Zeiss, Lightsheet Z.1). We

selected a few planes to train Ilastik to detect either DAPI or DsRed-positive neurons. Then the cell density in the whole telencephalon.

## Brain registration

To compare fluorescent signals among multiple brain samples, we performed image registration with CMTK registration GUI in Fiji (*Ostrovsky et al., 2013*); for details, please see https://github.com/jefferislab/BridgingRegistrations (*Jefferis Lab, 2015*) as described in a previous report (*Ito et al., 2013*). To make a reference brain, we used the DAPI signals of the best images obtained by light-sheet microscopy and processed with Gaussian blur. Three-dimensional reconstruction images were made with FluoRender (*Wan et al., 2012*).

## Clonal units identification

To identify the structure of clonal units, first we compared the structure of GFP-positive neurons among the registered brains in FluoRender. We detected 523 GFP-positive neural populations from 81 fish. We extracted GFP-positive clusters as clonal units when we found GFP-positive regions over-lapped in multiple registered brains. We tracked GFP-positive neuronal projections by observing both registered brains and raw images. We found fewer GFP-positive populations in Dm2b and Dm3a. Since we gave mild heat shock to induce Cre-lox recombination by thermal cycler, it could be possible that the embryos tend to stay in a certain posture in the PCR tubes so that a few neural stem cells are not likely to get heat shocked, which led to this bias. Also, as shown in *Figure 1—figure supplement 2B*, the labeling by HuC promoter is not homogeneous; the labeled signals are especially weak in the anterior part of the pallium (Dla). We think that is why we did not observe the clones in Dla in a reproducible way. So, we removed the clones in Dla from the comprehensive structural analysis. In *Figure 2B*, we showed cell lineages in Dla that were widely labeled.

## Immunohistochemistry and in situ hybridization

In order to define the anatomical regions, we examined the expression of marker genes by immunos-taining as previously reported (*Okuyama et al., 2013*). Briefly, after cutting 14-um-thick cryosections of adult brains, sections were incubated overnight with the primary antibody (anti-CaMK2α [Abcam, ab22609], parvalbumin [Millipore, MAB1572], GAD65/67 [Sigma, G5163], and GAD 67 [Sigma G5038], 1:2000 each), synapsin (Synaptic Systems, 106 011), and PSD (Abcam, ab18258) in 1% BSA/DMSO/Triton-X1000/PBS. Though we also tried anti-GAD 65 so far, we could not find the difference of inten-sity among anatomical regions. In order to visualize the distribution of radial glial cells, we used the anti-GFAP antibody (Abcam, ab7260). After washing several times with DMSO/Triton-X1000/PBS, the sections were incubated with fluorescent Alexa 488 (Thermo Fisher, 1:1000) and DAPI (1:4000) in 1% BSA/DMSO/Triton-X1000/PBS.

To analyze the expression of genes in the pallium, we performed ISH as previously reported (*Isoe et al., 2012*). The primers designed for making RNA probes are listed below:

| gene_F (forward) or R (reverse) | primer |
| --- | --- |
| c1ql3b_F | TATGAGATGTTGGGCACCTGT |
| c1ql3b_R | TGACCGTTCTTGCAGAGATCT |
| egr1_F | TGCAACCAAGACCGAGATGAT |
| egr1_R | AGCTGCGACTGATAGACACCT |
| eomesa_F | TGTGATGAATTGATAGGGGAA |
| eomesa_R | AAGTTCGTCTATGTTGTACCGT |
| fosl2_F | TATGGCCATCATCTGACCAA |
| fosl2_R | TGTGCTTCAAAAACACAGGA |
| lhx2_F | TAACGAGAATGACGGCGAGT |

*Continued on next page*

*Continued*

| gene_F (forward) or R (reverse) | primer |
| --- | --- |
| lhx2_R | AAGTCTGAAGGTGTGAACAGT |
| lhx6_F | AAAACCTCAAACGAGCAGCT |
| lhx6_R | TGACTCCTGCGCAATCTTTGA |
| neurod1_F | ATCTGACGGACAGCGATCCA |
| neurod1_R | TTAGTGGTTGGCTGAGACAGA |
| npas4_F | TCGTCTCTTCCGCTGTCGTTT |
| npas4_R | TCCTCCTCCAGAGAGGAGAT |
| pcdh9_F | TGGTGGCAGTCCACAGAAAT |
| pcdh9_R | TCCAGCAGATACTGGTTGTCAT |
| rfx4_F | TTCATCATGATGTACAGAACACA |
| rfx4_R | ACGCCTGACATAGATGGTTTAA |
| tbr1_F | ACGGATGTTTCCCTTTTTGA |
| tbr1_R | AGCCTGTGTAGACAGTGTCAT |
| TSHZ1_F | ATGCCCGAGGATGAGCTCAA |
| TSHZ1_R | ACGTTGCTTGTGTGGTTGACA |
| znf827_F | TCTTTGGAACTTCTTGGCTTGGAT |
| znf827_R | TGACACAAGCGAAAGAAAGGAAT |

## Brain slicing for ATAC-seq and RNA-seq

Adult medaka fish of Tg (HuC:DsRed) were anesthetized using tricaine methane sulfonate (MS222) solution (0.4%). After brain dissection, brains were immersed into artificial cerebrospinal fluid (ACSF) on ice. Then, brains were embedded in 2.5% low-melting agarose in ACSF and frozen at –20°C for 8 min. After removing the excess agarose gel, 130-µm-thick brain slices were cut using a vibrating blade microtome (Leica VT1000S). The slices were collected, placed onto the slide glass, and dissected using a razor blade under a fluorescence microscope based on the DsRed or GFP signal (Leica MZ16F).

## Randomization and blinding in sequencing analysis

We had one investigator who committed to collecting all biological samples for ATAC-seq and RNA-seq, and we had another investigator who performed the extraction of DNA or RNA randomly, performed sequencing analysis, and was blinded to the brain regions where samples were derived from.

## ATAC-seq

ATAC-seq was performed as previously described (*Buenrostro et al., 2013*) with some modifications. After collecting the GFP-positive compartments into a 1.5 ml tube with PBS, the nuclei were extracted in 500 µl of cold lysis buffer (10 mM Tris-HCl pH 7.4, 10 mM NaCl, 3 mM MgCl$_2$, 0.1% Igepal CA-630), centrifuged for 10 min at 500 × *g*, and supernatant was removed. Tagmentation reaction was performed as described previously (*Buenrostro et al., 2013*) with Nextera Sample Preparation Kit (Illumina). After tagmented DNA was purified using MinElute kit (QIAGEN), two sequential PCR were performed to enrich small DNA fragments. First, nine-cycle PCRs were performed using indexed primers from Nextera Index Kit (Illumina) and KAPA HiFi HotStart ReadyMix (KAPA Biosystems), and amplified DNA was size selected to a size of less than 500 bp using AMPure XP beads

(Beckman Coulter). Then a second seven-cycle PCR was performed using the same primer as the first PCR and purified by AMPure XP beads. Libraries were sequenced using the Illumina HiSeq 1500 platform.

## RNA-seq

After slicing the Tg (HuC:DsRed) brain using a vibratome, the pallium, subpallium, and Dd regions were dissected based on the DsRed signals and collected in Trizol (Thermo Fisher Scientific). 30 fish were used for one replicate and two replicates were made. Brain slices were homogenized in 1 ml of Trizol, and 200 μl of chloroform was added, and total RNA was isolated using RNeasy MinElute Cleanup Kit (QIAGEN). mRNA was enriched by poly-A capturing and RNA-seq libraries were generated using KAPA mRNA HyperPrep Kit (KAPA Biosystems). Libraries were sequenced using the Illumina HiSeq 1500 platform.

## ATAC-seq data processing

The sequenced reads were preprocessed to remove low-quality bases and adapter derived sequences using Trimmomatic v0.32 (*Bolger et al., 2014*), and then aligned to the medaka reference genome version 2.2.4 (ASM223467v1) by BWA (*Li and Durbin, 2009*). Reads with mapping quality (MAPQ) ≥ 20 were used for the further analyses. MACS2 (version 2.1.1.20160309; *Zhang et al., 2008*) was used to call peaks and generate signals per million reads tracks using the following options; ATAC-seq: macs2 callpeak --nomodel --extsize 200 --shift –100 -g 600000000 -q 0.05 -B --SPMR.

We evaluated the quality of our sequenced samples by the following methods and used only high-quality ATAC-seq data. First, we counted the number of mapped reads after removing redundant reads and selected samples with 2 million reads or more. Then, we calculated the fraction of reads in peaks (FRiP) values (*Landt et al., 2012*) and selected the data that have a FRiP value of 0.2 or higher.

We collected all peaks from 65 ATAC-seq data that met the above criteria and merged the peaks if there were overlaps between them. For each peak region, the number of mapped reads per total reads was calculated for each sample and then normalized by the average of all samples.

Hierarchical clustering was performed by calculating a Euclidean distance matrix and applying Ward clustering (*Ward, 1963*) using the *hclust* and *dendrogram* functions in R. ATAC-seq peaks were clustered using k-means clustering using *kmeans* function in R. PCA was performed using the *prcomp* function in R. t-distributed stochastic neighbor embedding (tSNE) (*Van Maaten & Hinton, 2008*) plot was generated using *Rtsne* package in R, with perplexity = 5 option. Uniform Manifold Approximation and Projection (UMAP) (*McInnes et al., 2018*) was performed using *umap* package in R, with n_neighbors = 7 and n_components = 2 options.

To identify the target gene of each ATAC-seq peak, the closest transcription start site (TSS) was searched and determined as the target gene if the distance to the TSS was less than 10 kb.

## RNA-seq data processing

The sequenced reads were preprocessed to remove low-quality bases and adapter derived sequences using Trimmomatic v0.32 (*Bolger et al., 2014*) and were aligned to the medaka reference genome version 2.2.4 (ASM223467v1) by STAR (*Dobin et al., 2013*), and reads with mapping quality (MAPQ) ≥ 20 were used for the further analyses.

Genes expressing significantly higher or lower in Dd2 were identified using DESeq2 (padj<0.05) (*Love et al., 2014*).

## GO analysis

GO analyses were performed using the GO resource (*Gene Ontology Consortium, 2021*).

## TF motif identification

MotifsGenome.pl script from HOMER program (*Heinz et al., 2010*) was used to identify the enriched motifs at ATAC-seq peaks.

## Comparison between medaka and human ATAC-seq data

We used region-specific (neocortex, primary visual cortex, hippocampus, mediodorsal thalamus, and striatum) OCRs from human brain ATAC-seq data (*Fullard et al., 2018*). Enrichments of known motifs

were compared between medaka OCR clusters and human region-specific OCRs, and the number of shared motifs was counted. The annotatePeaks.pl script from HOMER program (*Heinz et al., 2010*) was used to examine the enrichment of medaka OCR C6 motifs 1–3 in human region-specific OCRs with -size 2000 -hist 20 options.

## Analysis of gene expression in the mammalian brain

We examined the expression levels of medaka Dd2-preferentially expressed genes (809 genes) using the ISH data of adult mouse in Allen Brain Atlas (https://mouse.brain-map.org/). First, we downloaded normalized gene expression levels (expression energy of structure unionizes) for all genes from the Allen Brain Atlas. Next, we obtained the medaka orthologs of those genes using Ensembl's Biomart. We excluded cases where multiple medaka genes corresponded to a single mouse gene. To correct for expression bias across tissues caused by ISH, we normalized the expression levels of genes preferentially expressed in medaka's Dd2 in each tissue to the average expression level of all genes.

## Acknowledgements

We are grateful to many thoughtful comments and heartfelt support from Takeo Kubo at University of Tokyo. We thank Florian Engert at Harvard University for reading and editing the manuscript. We acknowledge comments from Kei Ito at University of Cologne and Nikolai Hoermann at Harvard University. We appreciate the financial and experimental support from the Takeo Kubo lab at University of Tokyo, the Florian Engert lab, Nicholas Bellono lab, Sophia Liang in Catherine Dulac lab, and Burcu Erdogan in Jessica Whited lab at Harvard University. This work was supported by the NIBB Collaborative Research Program (16-522, 17-513, and 18-513) to H Takeuchi; Japan Society for the Promotion of Science (JSPS) KAKENHI grant numbers JP 16H06987 (to YI), 16KT0072 (to HTakeuchi), 20K20303 (to YK); Japan Society for the Promotion of Science (JSPS) grant number JP21K06013 and Grant-in-Aid for Scientific Research on Innovative Areas grant number JP21H00245 to RN.

## Additional information

### Funding

| Funder | Grant reference number | Author |
|---|---|---|
| National Institute for Basic Biology | NIBB Collaborative Research Program 16-522 | Hideaki Takeuchi |
| Japan Society for the Promotion of Science | KAKENHI 16H06987 | Yasuko Isoe |
| Japan Society for the Promotion of Science | KAKENHI 16KT0072 | Hideaki Takeuchi |
| Japan Society for the Promotion of Science | KAKENHI 20K20303 | Yasuhiro Kamei |
| Japan Society for the Promotion of Science | JP21K06013 | Ryohei Nakamura |
| Japan Society for the Promotion of Science | Grant-in-Aid for Scientific Research on Innovative Areas JP21H00245 | Ryohei Nakamura |
| National Institute for Basic Biology | 18-513 | Hideaki Takeuchi |
| National Institute for Basic Biology | 17-513 | Hideaki Takeuchi |

The funders had no role in study design, data collection and interpretation, or the decision to submit the work for publication.

## Author contributions
Yasuko Isoe, Conceptualization, Data curation, Formal analysis, Funding acquisition, Investigation, Visualization, Methodology, Writing - original draft, Writing – review and editing; Ryohei Nakamura, Conceptualization, Data curation, Formal analysis, Funding acquisition, Investigation, Visualization, Methodology, Writing – review and editing; Shigenori Nonaka, Yasuhiro Kamei, Methodology, Writing – review and editing; Teruhiro Okuyama, Writing – review and editing; Naoyuki Yamamoto, Formal analysis, Methodology, Writing – review and editing; Hideaki Takeuchi, Funding acquisition, Writing – review and editing; Hiroyuki Takeda, Supervision, Writing – review and editing

## Author ORCIDs
Yasuko Isoe (iD) http://orcid.org/0000-0002-1391-7921
Ryohei Nakamura (iD) http://orcid.org/0000-0002-7812-8034
Yasuhiro Kamei (iD) http://orcid.org/0000-0001-6382-1365
Teruhiro Okuyama (iD) http://orcid.org/0000-0003-1566-0063

## Ethics
All experiments were conducted using protocols specifically approved by the Animal Care and Use Committee of the University of Tokyo (permit number: 12-07). All surgeries were performed under anesthesia using MS-222, and all efforts were made to minimize suffering, following the NIH Guide for the Care and Use of Laboratory Animals.

## Decision letter and Author response
Decision letter https://doi.org/10.7554/eLife.85093.sa1
Author response https://doi.org/10.7554/eLife.85093.sa2

# Additional files

## Supplementary files
• MDAR checklist

## Data availability
Sequencing data generated in this study have been submitted to the DDBJ BioProject database under accession number PRJDB14398. Imaging data generated for clonal structural analysis in this study and a table describing each visualized clone that was used for the statistics have been submitted to Dryad Digital Repository (https://doi.org/10.5061/dryad.ttdz08m2s).

The following datasets were generated:

| Author(s) | Year | Dataset title | Dataset URL | Database and Identifier |
|---|---|---|---|---|
| Isoe Y, Nakamura R, Nonaka S, Kamei Y, Okuyama T, Yamamoto N, Takeuchi H, Takeda H | 2022 | Epigenetically distinct synaptic architecture in clonal compartments in the teleostean dorsal pallium | https://ddbj.nig.ac.jp/ resource/bioproject/ PRJDB14398 | DDBJ BioProject, PRJDB14398 |
| Isoe Y | 2023 | Epigenetically distinct synaptic architecture in clonal compartments in the teleostean dorsal pallium | https://dx.doi.org/10. 5061/dryad.ttdz08m2s | Dryad Digital Repository, 10.5061/dryad.ttdz08m2s |

The following previously published dataset was used:

| Author(s) | Year | Dataset title | Dataset URL | Database and Identifier |
|---|---|---|---|---|
| Fullard JF, Hauberg ME, Bendl J, Egervari G, Cirnaru MD, Reach SM, Motl J, Ehrlich ME, Hurd YL, Roussos P | 2018 | An Atlas of Chromatin Accessibility in the Adult Human Brain | https://www.ncbi. nlm.nih.gov/geo/ query/acc.cgi?acc= GSE96949 | NCBI Gene Expression Omnibus, GSE96949 |

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
