## [Editor Report]

This important article highlights the clonal organization of the dorsal telencephalon, a major region of the vertebrate brain. The authors’ analyses reveal a distinct gene expression and provide a high-quality chromatin accessibility map of the adult teleost fish medaka. In addition, synaptic genes have a distinct chromatin landscape and expression pattern from one of the regions of the dorsal pallium, aiming to describe an evolutionary origin for these neurons.

---

## [Decision Letter]

**Decision letter after peer review:**

Thank you for submitting your article "Epigenetically distinct synaptic architecture in clonal compartments in the teleostean dorsal pallium" for consideration by *eLife*. Your article has been reviewed by 3 peer reviewers, and the evaluation has been overseen by a Reviewing Editor and Marianne Bronner as the Senior Editor. The reviewers have opted to remain anonymous.

Essential revisions:

1. Please try to better integrate and discuss the massive amount of high-quality data generated. Presented information in many cases is difficult to digest and several suggestions have been listed by the reviewers to address this point.

2. Data supporting the differential expression of synaptic genes is not fully presented. It would be good to show all differentially expressing genes.

3. The comparison between medaka fish and human telencephalon regions would benefit from a more extensive molecular analysis.

4. It would be good to assess what percentage of neurons within one domain are being targeted to have an estimate of how representative the clonal and connectivity information is. Also, higher-resolution images would be beneficial to exemplify the projections described.

*Reviewer #1 (Recommendations for the authors):*

Here are specific comments on the paper that might help the authors clarify some of the important concepts that they aim at sharing.

– Introduction, Page 3, lines 68 – 71. The authors refer to "cellular sub-populations" and "clonal units" as if these were synonyms. There is some confusion in these lines about the difference between lineages and cells with similar functions. Even if the authors want to make the point that clonal history and cell types are linked in the dorsal telencephalon, it is important to stress in the introduction that this is not the universal case – quite the opposite indeed in most stem cell systems.

– Introduction, Page 4, lines 92-93. The claim "We propose that changes in the clonal units in the pallium is the driving force for the anatomical diversity during evolution." is not funded by experimental data from other species. The authors should share with the reader what pieces of evidence they have collected to make such a statement.

– Results, Page 7, lines 152-3. Also, Discussion, page 17, lines 361-3. The authors use here and elsewhere the term "mutually exclusive" when talking about the clones in the pallium. What does this mean? Isn't any clone mutually exclusive with another clone, by definition? The authors could re-phrase this to make their statement clear. This is about clones contributing exclusively to one region and not to another. If the authors develop this concept further, it would be appropriate to discuss this finding in relation to how boundaries are established in biological systems – i.e., early works in *Drosophila* where clones do not cross segments, on the mammalian intestine where clones do not cross villi territories, on the medaka retina and on the zebrafish heart where clones define sectors.

– Results, Page 8, Section "Clonal Units tend to project to the same target regions". I do not see the added value for this section. Projections in the brain of different organisms of the same species follow the same pattern and do look alike. The authors had just stated that the anatomical regions are indeed clonal units. It is therefore natural that they project to the same targets. The authors should be more explicit in case they want to communicate something different.

– Discussion, page 18, lines 377 – 383. The authors are using a common regulatory network as evidence for a common evolutionary origin. This can not be the only speculation, mostly in a manuscript that highlights organization in clonal sectors. Can the authors expand their discussion integrating the clonal aspect that they have characterized?

– Discussion, page 19, lines 442 – 444. The authors refer to "synaptic architecture" in the pallium. I can not find any data in the paper that assess the synaptic architecture. There is no staining, no EM data, no functional output, and no visualization of how a synapsis looks like in the medaka pallium. The authors have indeed done much to characterize the genome architecture and identified synaptic genes in their bulk analysis. Without the characterization of synapsis in Dd2 neurons, the message of this paragraph needs to be toned down.

– "The pallium in teleosts has been mysterious, since different species exhibit, within its boundaries, different numbers or cellular densities of compartmentalized anatomical sub-regions". Can the authors explain this concept further? They are making a big point with the comparison between the pallium and the subpallium, whose relevance is not clear to the non-expert reader.

– In mammals, neurons generated in the sub-pallium migrate towards the pallium. And in fish? Is this happening? Is this the reason the authors opted for a HuC promoter and not a ubiquitous promoter? If it is the case, the authors should become clear about it, and discuss this fact openly.

*Reviewer #2 (Recommendations for the authors):*

1. Atlas: Nomenclature should be more accessible to readers. An easy-to-access table with the abbreviations and their meanings could be incorporated. Additionally, a succinct comparison (in the paper body) with the previous atlas might be helpful to underscore the differences and new information provided by this atlas. The clonal analysis could be integrated with the anatomical atlas to further define pallial regions. The authors should consider subdividing some of the regions based on the clones distribution they found, such as 3 clonal units in Dm2 and Dm3, 5 clonal units in Dld, 2 layers in Dlv, and 2 regions with distinct projections in Dp.

2. Table 2 (and supplementary figure 3) provides a wealth of information, but is very difficult to digest. Could the authors think of better ways of digesting and presenting the information? For instance, how many clones per animal? How many animals had clones in dorsal or ventral pallium or both? How many animals had clones in one vs more than one dorsal pallium region? How many clones had intermingled green and red cells in dorsal vs ventral pallium? Etc. Also true for summarizing and presenting connectivity data.

3. The connectivity section of the paper is underexplored. As said in point 2, this information could be incorporated to better refine the atlas. Higher-resolution images would be beneficial to exemplify the projections described. The connection matrix shows the number of fishes with a given projection. Is this resolved clearly in fishes with more than one clonal unit? Given that the number of clones per region is not homogenous, these values should be normalized. What is the implication of the findings? Speculations behind a few cross-boundary connections? Comparison to other teleost species?

4. Dd2 doesn't seem to be the only region containing a specific set of OCR. Are there other processes enriched in the OCR of other domains? Can this be used to identify genes selectively expressed in different domains? Could transcription factors associated with other domains also be identified? Besides, based on the pie charts, it seems that rather than being a Dd2 exclusive characteristic, all region-specific OCR clusters correspond mostly to intronic or intergenic regions and are likely to be enhancers. (line 222)

5. Data supporting the differential expression of synaptic genes is not presented, but only shown as a summary cartoon in figure 4d and supplementary figure 7. It would be convenient to show the differential expression of the synaptic genes and subunits of receptors, especially to support the statement in line 285 that inhibitory synaptic genes are specifically downregulated.

6. I'm curious to see what other genes (besides synaptic genes) are differentially expressed or differentially accessible. Can this information be used to further substantiate the search for homology between teleost and mammalian pallium (figure 5e)? Statement in line 349 about similar transcriptional regulators would benefit from additional support. The homology between human and medaka pallial regions is based on the motif binding site of 2 transcription factors, and requires more solid evidence.

7. HucD labels progenitors and immature neurons, while this is addressed by the authors in the discussion, it would be convenient to assess what percentage of neurons within one domain are being targeted to have an estimate of how representative the clonal and connectivity information is. Does a different driver generate similar results? Is the clonal compartmentalization retained when neurons reach maturity?

8. The atlas and clonal information are very valuable to others working in medaka or teleost telencephalon, and while it is appropriate to distribute the data through a public repository, the authors could consider an interactive webpage for exploration.

*Reviewer #3 (Recommendations for the authors):*

My main concern is about the data interpretation of the clonal tracing experiment:

In the study of the clonal architecture of the medaka brain, the authors reached the conclusion of "the anatomical regions in the pallium are formed by mutually exclusive compartment of clonally-relate cell bodies" based on the evidence of 1) GFP-signal distribution is different in pallium (exclusive) and subpallium (mixed green and red cells) regions; and 2) different "clonal unit" repeatedly occur in different subregions of the pallium. I would like to point out that this conclusion relies on two assumptions: a) Every individual fish follows the same pattern of neurogenesis, and b) common progenitor cells only give rise to cells (the so-called "clonal units") within the same region within pallium. Given the regulatory developmental process of vertebrates and the fact certain progenitors can migrate far after being born in the brain, neither of the assumptions may hold true.

One experiment the authors should add is to induce the heat-shock at both earlier and later time points. If the "clonal unit" truly reflects lineally related cells, they should only co-occur in a later-induced clone but not in early-induced clones. Changing the timing of heat-shock will also give the authors a hint of how "unique" the pallium is in terms of clonal compartmentation in contrast to the subpallium region. For example, the subpallium is also compartmentalized, but at a later time point.

Otherwise, the paper is very well written. A possible typo on Page 17. Line 371: "…the distribution of cells in clonal cells". The authors may have meant "clonal units"?

---

## [Author Response]

Essential revisions:1. Please try to better integrate and discuss the massive amount of high-quality data generated. Presented information in many cases is difficult to digest and several suggestions have been listed by the reviewers to address this point.

In a revised manuscript, we performed several new analyses to integrate and interpret the data we acquired for the structural analysis of clonal-units. We appreciated the reviewers’ comments and tried to answer them.

2. Data supporting the differential expression of synaptic genes is not fully presented. It would be good to show all differentially expressing genes.

We presented the RNA-seq data of all synaptic genes that express differently in the medaka dorsal pallium (Dd2) in the revised manuscript.

3. The comparison between medaka fish and human telencephalon regions would benefit from a more extensive molecular analysis.

In our revised manuscript, we added an analysis of the gene expression level across the mammalian brain regions to compare the transcriptional profiles between medaka Dd2 and various brain regions in mice. As a result, we could not observe strong correlations with any specific brain regions in mice. Therefore, we have revised our conclusions regarding the correspondence between medaka's Dd2 and mammalian brain regions to be more cautious.

4. It would be good to assess what percentage of neurons within one domain are being targeted to have an estimate of how representative the clonal and connectivity information is. Also, higher-resolution images would be beneficial to exemplify the projections described.

We assess the ratio of neurons and DsRed-positive cells by staining. Also, we added higher-resolution images in the revised manuscript figures.

Reviewer #1 (Recommendations for the authors):Here are specific comments on the paper that might help the authors clarify some of the important concepts that they aim at sharing.– Introduction, Page 3, lines 68 – 71. The authors refer to "cellular sub-populations" and "clonal units" as if these were synonyms. There is some confusion in these lines about the difference between lineages and cells with similar functions. Even if the authors want to make the point that clonal history and cell types are linked in the dorsal telencephalon, it is important to stress in the introduction that this is not the universal case – quite the opposite indeed in most stem cell systems.

Thank you so much for your comment. Here is the revised version:

“First, cell-lineage analysis was used to visualize cellular subpopulations derived from the same neural stem cells at an early developmental stage. Since post-hatch neurogenesis occurs throughout life in all teleosts, such cell lineage analysis of neural progenitors allows us to collectively label individual neural progenitors populations that may serve similar functions” (page 3, line 68-71)

– Introduction, Page 4, lines 92-93. The claim "We propose that changes in the clonal units in the pallium is the driving force for the anatomical diversity during evolution." is not funded by experimental data from other species. The authors should share with the reader what pieces of evidence they have collected to make such a statement.

Thank you for the comment. In our revised manuscript, we removed this sentence from the Introduction, but we added sentences to discuss the relationship between clonal units and evolutionary diversity, which include the following discussion.

– Results, Page 7, lines 152-3. Also, Discussion, page 17, lines 361-3. The authors use here and elsewhere the term "mutually exclusive" when talking about the clones in the pallium. What does this mean? Isn't any clone mutually exclusive with another clone, by definition? The authors could re-phrase this to make their statement clear. This is about clones contributing exclusively to one region and not to another.

Thank you for your comment. As the reviewer comment says, we should have described that “in the pallium, we suggest that cell lineages distribute exclusively to one anatomical region and not to another”, which is added in the revised manuscript. (page 8, line 172)

If the authors develop this concept further, it would be appropriate to discuss this finding in relation to how boundaries are established in biological systems – i.e., early works in Drosophila where clones do not cross segments, on the mammalian intestine where clones do not cross villi territories, on the medaka retina and on the zebrafish heart where clones define sectors.

Thank you for the suggestion. We added a paragraph in the discussion about the how boundaries are established in biological systems:

“In animal bodies, various boundaries are formed by clonally related cells. In the *Drosophila* brain, clones do not cross segments, suggesting that clones are projection units ^29^. In the zebrafish heart ^34^, as fish grow from juveniles to adults, the whole heart is formed by cells derived from a small number (~8) of cardiomyocytes, which define the sections through clonal dominance. In the mammalian gastrointestinal tract^76^, several organs, such as the esophagus and glandular stomach, are formed by patches of clonal units that exhibit sharp boundaries and serve distinct digestive functions. Different expression patterns of *Hox-*cluster genes contribute to the boundary maintenance. In the medaka retina ^35^, retinal stem cells possess multipotency and generate all neuron types, and clonally-related cells form the cylindrical structure ^36^.

We speculate that clonal boundaries in the medaka pallium have been established with the similar mechanism as in medaka’s retinal boundaries. In the adult teleost pallium, the radial glia cells on the surface of the telencephalic hemisphere, which project toward the inside of the hemisphere, function as neural stem cells and generate neural progenitors throughout life (Figure 1 —figure supplement 4). This cellular arrangement might contribute to the compartmentalized structure.” (page 22, line 419-431)

– Results, Page 8, Section "Clonal Units tend to project to the same target regions". I do not see the added value for this section. Projections in the brain of different organisms of the same species follow the same pattern and do look alike. The authors had just stated that the anatomical regions are indeed clonal units. It is therefore natural that they project to the same targets. The authors should be more explicit in case they want to communicate something different.

Thank you for the comment. It is not always true that cells whose cell bodies are located in the same region project to the same target with long axonal bundles. For example, the cell bodies in the subpallium are located inside the boundaries, but we found that the projections were mixed and don’t form axon bundles like the clonal units in the pallium. “Based on this observation that cell lineages in the pallium tend to project to the same target, we concluded that the labeled cellular sub-populations are clonal units.” (page 10, line 208-210) In the revised manuscript, we took more care in distinguishing “cell lineages'' and “clonal units.”

– Discussion, page 18, lines 377 – 383. The authors are using a common regulatory network as evidence for a common evolutionary origin. This can not be the only speculation, mostly in a manuscript that highlights organization in clonal sectors. Can the authors expand their discussion integrating the clonal aspect that they have characterized?

Thank you for the comment. Here we added a few sentences in the revised manuscript,

“In addition, our PCA analysis on ATAC-seq data with clonal units show that open chromatin patterns in medaka Dd are close to that in Dl. Taken together with the modular nature of clonal units in the pallium, it is likely that Dd has been differentiated from the neighboring anatomical region, Dl.” (page 23, line 459-462)

– Discussion, page 19, lines 442 – 444. The authors refer to "synaptic architecture" in the pallium. I can not find any data in the paper that assess the synaptic architecture. There is no staining, no EM data, no functional output, and no visualization of how a synapsis looks like in the medaka pallium. The authors have indeed done much to characterize the genome architecture and identified synaptic genes in their bulk analysis. Without the characterization of synapsis in Dd2 neurons, the message of this paragraph needs to be toned down.

Thank you for your comment. In our figure 4 and the supplemental figure, we stained synaptic genes, such as synapsin and PSD95, but yes, we don’t have more detailed architectural data. So here we used “synaptic properties” instead of “synaptic architecture”. (page 26, line 524)

– "The pallium in teleosts has been mysterious, since different species exhibit, within its boundaries, different numbers or cellular densities of compartmentalized anatomical sub-regions". Can the authors explain this concept further? They are making a big point with the comparison between the pallium and the subpallium, whose relevance is not clear to the non-expert reader.

Thank you for your comment. In our revised manuscript, we updated as follows, “The pallium in teleosts has been mysterious, since different species exhibit, within its boundaries, different numbers or cellular densities of compartmentalized anatomical sub-regions. On the other hand, the subpallial regions are relatively conserved among teleost species in terms of anatomical sub-regions and gene expressions (Moreno et al. 2009; Northcutt and Glenn Northcutt 2008).” (page 23, line 442-445)

– In mammals, neurons generated in the sub-pallium migrate towards the pallium. And in fish? Is this happening? Is this the reason the authors opted for a HuC promoter and not a ubiquitous promoter? If it is the case, the authors should become clear about it, and discuss this fact openly.

Different from mammals, neurons generated in the sub-pallium don’t dynamically migrate inside the pallial regions in teleosts (Grandel et al. 2006; Labusch et al. 2020; Ganz et al. 2010).

Reviewer #2 (Recommendations for the authors):1. Atlas: Nomenclature should be more accessible to readers. An easy-to-access table with the abbreviations and their meanings could be incorporated.

Thank you for the helpful comment. We made a table with abbreviations and their meanings as Table 2. (page 36, Table 2)

Additionally, a succinct comparison (in the paper body) with the previous atlas might be helpful to underscore the differences and new information provided by this atlas.

Thank you for the comment. We added a few sentences in the revised manuscripts, “Here, we redefined the medial part of the pallial region (Dm) into several sub compartments (Dm2a, Dm2b, Dm3a, and Dm3b) by clear DAPI-positive boundaries. These sub compartments are absent in previous brain atlases.”. (page 6, line 133-135)

The clonal analysis could be integrated with the anatomical atlas to further define pallial regions. The authors should consider subdividing some of the regions based on the clones distribution they found, such as 3 clonal units in Dm2 and Dm3, 5 clonal units in Dld, 2 layers in Dlv, and 2 regions with distinct projections in Dp.

Thank you for the comment. Though we considered updating the existing pallial regions with sub-regions based on clonal unit structure, we didn’t update the anatomical atlas since we only could visualize this clonal unit structure by lineage labeling. However, our clonal ATAC-seq revealed the differential open-chromatin pattern among clonal units in Dlv (Figure 2D, Dld_mv vs Dld_pv), which suggest that there are actually different cellular populations in the anatomical regions. In the revised manuscript, we added a new panel in Figure 7AB, so that we can highlight this finding. (page 21, Figure 7A-C, line 406-408)

2. Table 2 (and supplementary figure 3) provides a wealth of information, but is very difficult to digest. Could the authors think of better ways of digesting and presenting the information? For instance, how many clones per animal? How many animals had clones in dorsal or ventral pallium or both? How many animals had clones in one vs more than one dorsal pallium region? How many clones had intermingled green and red cells in dorsal vs ventral pallium? Etc.

We decided registering the *old* Table 2 as an original data/resource (Dryad Digital Repository, doi:10.5061/dryad.ttdz08m2s), and we digested the information into some statistics and showed them in Figure 1 —figure supplement 3. In our experiments, we induced several (4-8 on average) clusters visualized per fish (Figure 1 —figure supplement 3 A, B). And we didn’t find strong asymmetricity in the induction of clonal units (Figure 1 —figure supplement 3B, D). In most of the fish, both the dorsal and ventral telencephalon had clones, but more clones in the pallium (Figure 1 —figure supplement 3 B, E). In the subpallium, most of the clones are not individually appeared but visualized with neighboring clones (Figure 1 —figure supplement 3 B), while we observed most of the clones in the pallium alone (Figure 1 —figure supplement 3 B,F). (page 7, line 159 – page 8, 167)

Also true for summarizing and presenting connectivity data.3. The connectivity section of the paper is underexplored. As said in point 2, this information could be incorporated to better refine the atlas.

We added a new panel as Figure 2B to show higher-resolution images and summary of the projection described. We found that most of the clones stay in the same region, but some clones project crossing other regions. (page 9, Figure 2B)

Higher-resolution images would be beneficial to exemplify the projections described.

We added a new panel as Figure 2B to show higher-resolution images and summary of the projection described. (page 9, Figure 2B)

The connection matrix shows the number of fishes with a given projection. Is this resolved clearly in fishes with more than one clonal unit? Given that the number of clones per region is not homogenous, these values should be normalized.

Thank you so much for the comment. We re-analyzed the projection data and normalized the projection value according to the reviewer’s comment (Figure 2A). We agree that this way is much better to show the actual probability of projection targets. (page 9, Figure 2A)

What is the implication of the findings? Speculations behind a few cross-boundary connections? Comparison to other teleost species?

Thank you for the comment. In the revised manuscript, we added a few lines to make our speculations clear: In the clonal structural analysis, we found that the cell bodies of the cell lineages in the pallium cluster together and locate inside the anatomical regions, but whether the projection of the cell lineages goes to the same target or not was not clear. After the projection analysis, we found that most cell lineages project to the same target, which implies that these cell lineages are also functional units. (page 10, line 205-207)

We found the clonal units in Dp project axons all the way to the olfactory bulb. This projection is conserved in teleost and experimentally visualized in zebrafish previously (Kermen et al. 2020). In the manuscript, we described that the expression of a few transcription factors (TFs) are conserved between the lateral pallium (Dl) of zebrafish and medaka (page 23, line 454-456). Though we haven’t analyzed the expression of TFs in Dp, we expect that there are some conserved expression of TFs in Dp of zebrafish and medaka.

4. Dd2 doesn't seem to be the only region containing a specific set of OCR. Are there other processes enriched in the OCR of other domains?

Thank you for the comment. We added a few sentences in the revised manuscript.

“Also, the GO terms related to early development were enriched in OCR cluster 14, and many metabolic and biosynthesis processes were enriched in OCR cluster 15. Also, one of the subpallial OCR clusters (OCR C5) was enriched with neuropeptide signaling, neural differentiation and vascular processes.” (page 14, line 277-280)

Can this be used to identify genes selectively expressed in different domains? Could transcription factors associated with other domains also be identified?

Thank you for the comment. In general, OCRs can function to both activate or repress gene expression, so RNA-seq data is required to identify genes specifically expressed in each domain. In this study, we chose Dd2 for RNA-seq analyses and found candidate TFs (page 18, Figure 6) In order to identify TFs associated with other brain regions, additional RNA-seq is required, and we leave it for a future study.

Besides, based on the pie charts, it seems that rather than being a Dd2 exclusive characteristic, all region-specific OCR clusters correspond mostly to intronic or intergenic regions and are likely to be enhancers. (line 222)

Thank you for the comment. We agree with this comment and we did not intend to claim that the OCR of Dd2 is unique in this matter. We added the following sentences, “We examined the distribution of ATAC-seq peaks in the genome, and Dd2-specific ATAC-seq peaks, located mainly in the intron and intergenic regions (Figure 3 —figure supplement 1D), suggesting that the genomic regions that exhibit these peaks function as Dd2-specific enhancers.” (page 12, line 251 – page 13, line 253)

5. Data supporting the differential expression of synaptic genes is not presented, but only shown as a summary cartoon in figure 4d and supplementary figure 7. It would be convenient to show the differential expression of the synaptic genes and subunits of receptors, especially to support the statement in line 285 that inhibitory synaptic genes are specifically downregulated.

In new Figure 5A, we added a heat map showing the all differential expression of synaptic genes which includes excitatory, inhibitory and modulatory synapses. We added the following sentences, “RNA-seq data showed differential expression of several synaptic genes in many types of neurons (modulatory, excitatory, and inhibitory neurons) (Figure 5A).” (page 16, Figure 5)

6. I'm curious to see what other genes (besides synaptic genes) are differentially expressed or differentially accessible. Can this information be used to further substantiate the search for homology between teleost and mammalian pallium (figure 5e)? Statement in line 349 about similar transcriptional regulators would benefit from additional support. The homology between human and medaka pallial regions is based on the motif binding site of 2 transcription factors, and requires more solid evidence.

Thank you for the comment. We modified Figure 4 to make it clearer which other genes are differentially expressed in Dd2 (See also Figure 4 —figure supplement 1). We showed that GO terms enriched with Dd2 OCR or genes are mainly synapse-related ones. Furthermore, we newly analyzed the expression of all Dd2-preferentially expressing genes (809 genes, which include non-synaptic genes) across the mouse brain regions (12 regions) using the Allen Institute dataset, but we couldn’t find the strong correlation with any specific brain regions in mice. We modified our manuscript to reflect these results. (page 20, line 396 – page 21, line 401)

7. HucD labels progenitors and immature neurons, while this is addressed by the authors in the discussion, it would be convenient to assess what percentage of neurons within one domain are being targeted to have an estimate of how representative the clonal and connectivity information is. Does a different driver generate similar results? Is the clonal compartmentalization retained when neurons reach maturity?

Thank you for the comment. We quantified the ratio of the labeled cells in the Tg line, we found that around 60-70% of cells in the telencephalon were labeled with DsRed in our transgenic line (Figure 1 —figure supplement 2B). (page 38, line 748-752)

We believe that we would get similar results in a different driver. In the previous paper (Dirian et al. 2014), clonal analysis was done in zebrafish using a ubiquitous promoter and showed a similar compartmentalized structure of cell lineages in the lateral pallium. Since the ubiquitous promoter is active in the mature neurons, we believe that the clonal compartmentalization is still retained with mature neurons.

8. The atlas and clonal information are very valuable to others working in medaka or teleost telencephalon, and while it is appropriate to distribute the data through a public repository, the authors could consider an interactive webpage for exploration.

Thank you for a good recommendation. Recently, as a different project, the first author is working on generating a larval medaka whole-brain atlas in an interactive website by extending the existing larval zebrafish whole-brain atlas (https://zebrafishatlas.zib.de/). We would include our medaka adult brain atlas to this atlas as well in the near future.

Reviewer #3 (Recommendations for the authors):My main concern is about the data interpretation of the clonal tracing experiment:In the study of the clonal architecture of the medaka brain, the authors reached the conclusion of "the anatomical regions in the pallium are formed by mutually exclusive compartment of clonally-relate cell bodies" based on the evidence of 1) GFP-signal distribution is different in pallium (exclusive) and subpallium (mixed green and red cells) regions; and 2) different "clonal unit" repeatedly occur in different subregions of the pallium. I would like to point out that this conclusion relies on two assumptions: a) Every individual fish follows the same pattern of neurogenesis, and b) common progenitor cells only give rise to cells (the so-called "clonal units") within the same region within pallium. Given the regulatory developmental process of vertebrates and the fact certain progenitors can migrate far after being born in the brain, neither of the assumptions may hold true.One experiment the authors should add is to induce the heat-shock at both earlier and later time points. If the "clonal unit" truly reflects lineally related cells, they should only co-occur in a later-induced clone but not in early-induced clones. Changing the timing of heat-shock will also give the authors a hint of how "unique" the pallium is in terms of clonal compartmentation in contrast to the subpallium region. For example, the subpallium is also compartmentalized, but at a later time point.

Thank you for the comment. As the reviewer commented, in order to see the difference of compartmentalization between the pallium and the subpallium, we should perform the lineage analysis experiment that recombination is induced at the earlier or later stage. In our previous paper (Okuyama et al. 2013), when we applied the induction at the earlier stage, we observed GFP expression in a wider area. And for the later-stage recombination-induction experiment, unfortunately, it was technically very difficult since the heat shock promoter in our transgenic lines we used shows much less activity in the later developmental stage (Okuyama et al. 2013), and we couldn’t induce the cre-loxp recombination with the mild heat shock (38 C for 15 min) we used for our experiments.

However, we still think we can respond back to the reviewer’s comment that is based on the two concerns:

I would like to point out that this conclusion relies on two assumptions: a) Every individual fish follows the same pattern of neurogenesis, b) common progenitor cells only give rise to cells (the so-called "clonal units") within the same region within the pallium. Given the regulatory developmental process of vertebrates and the fact certain progenitors can migrate far after being born in the brain, neither of the assumptions may hold true.

We also agree that the two assumptions are not true.

However, the image in the figure 1D is not made from multiple fish, but the image is a raw data from a single fish. So GFP-positive cells and DsRed-positive cells are actually mixed in the subpallium, but not in the pallium, even in individual fish. (page 6, line 111-112)

Also, in teleosts, adult neurogenesis occurs proficiently in the telencephalon but in a different structural mechanism in the pallium and subpallium (Grandel et al. 2006), (Labusch et al. 2020; Ganz et al. 2010). In the pallium, the cell bodies of radial glial cells (neural stem cells) are located on the surface of the hemisphere project toward the inside of the hemisphere (Figure 1 —figure supplement 4). And neural progenitors derived from the neural stem cells migrate along this projection. Thus we assume the pallial clonal units visualized in our paper were derived from the neural stem cells that are located in the same region (Figure 1 —figure supplement 4). On the other hand, neural stem cells for the teleost subpallium locate the subpallial ventricle, and the neural progenitors migrate from there in the anterior-posterior axis (page 40, Figure 1 —figure supplement 4, line 772-781). Therefore, organization of radial glial cells is also clearly different between pallium and subpallium, which is consistent with the difference in compartmentalization property.

Taken together, although the induction of the cre-loxp recombination at the later stage is informative, the clonal structural differences between the pallium and subpallium is supported from the observation of individual fish and radial glial cell organization.

Otherwise, the paper is very well written. A possible typo on Page 17. Line 371: "…the distribution of cells in clonal cells". The authors may have meant "clonal units"?

Thank you so much. We’ve meant “clonal units” here, and this is corrected in the revised manuscript. (page 23, line 446)